# Brain Networks Should Be Learned, Not Constructed

Liang Yang [1]  Shuai Zhai [1]  Ziyi Ma [1]  Jiaming Zhuo [1]  Di Jin [2]  Chuan Wang [3*]  Zhen Wang [4]  Xiaochun Cao [5]

## Abstract

The construction of the brain functional network often follows the hand-crafted Correlation Coefficients without any learnable components. Meanwhile, most efforts are made to the models, such as graph neural networks, that make predictions with the constructed brain network as input. Unfortunately, the fixed brain network may lose critical information during construction and lead to difficulty in performance improvement, even with deliberately designed graph models. From this perspective, the current situation is similar to the machine learning community, *i.e.*, hand-crafted features and learnable predictors, before the advent of representation learning. In fact, the brain network can be regarded as a graph-structured learnable representation of the brain. By drawing on representation learning, this paper presents the Brain Representation (BRep) learning problem. To this end, the widely used linear and nonlinear correlations are enhanced to be high-order, parametric, and learnable. The expressive brain representation shifts the heavy computational burden away from downstream architectures, enabling a simple yet accurate predictor. Theoretical analysis guarantees the model's universal approximation to any U/V-statistics. Extensive evaluations demonstrate that BRep possesses superior performance, high efficiency, and interpretability. The code is available at https://github.com/Kevin-916/BRep-demo/.

[1]Hebei Province Key Laboratory of Big Data Calculation, School of Artificial Intelligence, Hebei University of Technology, Tianjin, China [2]Tianjin Key Laboratory of Cognitive Computing and Application, College of Intelligence and Computing, Tianjin University, Tianjin, China [3]School of Computer Science and Technology, Beijing JiaoTong University, Beijing, China [4]School of Artificial Intelligence, OPtics and ElectroNics (iOPEN), School of Cybersecurity, Northwestern Polytechnical University, Xi'an, China [5]School of Cyber Science and Technology, Shenzhen Campus of Sun Yat-sen University, Shenzhen, China. Correspondence to: Chuan Wang <wangchuan@bjtu.edu.cn>.

*Proceedings of the 43rd International Conference on Machine Learning*, Seoul, South Korea. PMLR 306, 2026. Copyright 2026 by the author(s).

## 1. Introduction

Research into brain's structure and function constitutes a central and invaluable endeavor in the life sciences, offering profound insights and serving as a critical guide for the diagnosis of neurological disorders. By characterizing the interactions among brain regions, brain networks offer a novel framework and powerful tool for advancing our understanding of the brain (Fornito et al., 2016).

The research on the brain network consists of three components: brain imaging (Tuan et al., 2025), brain network construction (Rubinov & Sporns, 2011), and brain network analysis (Bahrami et al., 2023). Brain imaging techniques can be broadly divided into structural imaging (sMRI (Fischl et al., 1999), DTI (Rubinov & Sporns, 2010), CT (He et al., 2007; Fleischer et al., 2019)), functional imaging (fMRI (Van Den Heuvel & Pol, 2010; Bielczyk et al., 2019), EEG (Rossini et al., 2019), MEG (Mandal et al., 2018), fNIRS (Li et al., 2022)), and molecular/metabolic imaging (PET (Veronese et al., 2019), SPECT (Imokawa et al., 2024), MRS (Soares & Law, 2009)). Brain network construction methods can be divided into functional connectivity (Van Den Heuvel & Pol, 2010), effective connectivity (Ray et al., 2021), and structural connectivity (Rubinov & Sporns, 2010). Brain network analysis often employs network/graph algorithms (Barabási, 2013; Foulds, 1995), including graph-theoretical measures (Rubinov & Sporns, 2010), community algorithms (Newman, 2006), graph kernels (Vishwanathan et al., 2010), GNNs (Kipf, 2016; Zhuo et al., 2023), graph self-supervised learning (Zhuo et al., 2024b; Xu et al., 2024; Zhuo et al., 2024a), and graph transformers (GTs) (Dwivedi & Bresson, 2020; Kreuzer et al., 2021; Zhuo et al., 2025a;b).

For the diagnosis of psychiatric and neurodegenerative disorders (Insel, 2010; Park & Friston, 2013), functional imaging (*e.g.*, fMRI) and functional connectivity, which describes the statistical dependencies between brain regions, are widely used. Specifically, the brain network is constructed from the hand-crafted Correlation Coefficients, *e.g.*, Pearson Correlation (Biswal et al., 1995), distance Correlation (Székely et al., 2007), and HSIC (Gretton et al., 2007), of BOLD (Ogawa et al., 1992). At the same time, most efforts are made on models that predict individual neurological disorders with the constructed brain network as input, and many flexible graph models, such as Graph Neural Networks

(GNNs) (Li et al., 2021; Cui et al., 2022; Kan et al., 2022a; Zhang et al., 2023) and Graph Transformers (GTs) (Kan et al., 2022b; Xu et al., 2024; Yu et al., 2024; Peng et al., 2025), are explored as shown in Fig. 1(a). Unfortunately, the fixed brain network may lose critical information during construction since simple and fixed correlation/dependence measures possess limited representation capability. Thus, it may lead to difficulty in performance improvement, even with deliberately designed graph models. The current situation is similar to the machine learning community, *i.e.*, hand-crafted features and learnable predictors, before the advent of representation learning (Bengio et al., 2013; Domingos, 2012; LeCun et al., 2015).

By drawing on representation learning, this paper presents the Brain Representation (BRep) learning problem. In this problem, the brain network is regarded as a graph-structured learnable representation of the brain, instead of the fixed input to the graph models. The representation should be flexible to capture comprehensive information and make the following predictor simple, *e.g.*, MLP, and accurate. To this end, the widely used linear and nonlinear correlations are enhanced to be high-order, parametric, and learnable. Specifically, linear and nonlinear correlations are unified as the inner product of the BOLD time series in high-dimensional latent spaces with *fixed* mappings, and the high-order dependence measure can be approximated with this inner product with a *learnable* mapping. The graph-structured representation and the parametric correlation estimator make the framework possess an end-to-end characteristic as shown in Fig. 1(b). Then, the framework is implemented by combining the parametric correlation and a TopK sparsification. The learning process is supervised by the individual's neurological disorder label and regularized by denoising the BOLD time series with the learned brain network. Finally, the universal approximations of the proposed methods, *i.e.*, bilinear functions with outer products and bilinear functions with multi-head inner products, to high-order U/V-statistics are theoretically analyzed. The main contributions of this paper are summarized as follows:

- We take the brain network as the representation of the brain, and present the problem of graph-structured brain representation learning.

- We propose a learnable, parametric high-order dependence measure by unifying and extending linear and nonlinear correlations.

- We implement a flexible brain representation learning method and simplify the downstream predictor to avoid the iterative message passing operator.

- We provide a theoretical analysis to show the universal approximation of the proposed model to U/V-statistics.

- We experimentally demonstrate that the proposed BRep is high-performance, scalable, and interpretable.

## 2. Graph-structured Representation Learning

As shown in Fig. 1(a), existing brain network modeling methods construct a brain network by employing different correlation coefficients, which are comprehensively developed in statistics without a learnable component. Thus, the fixed brain network may lose critical information during construction and prevent performance improvement in the following graph model, since simple dependence measures possess limited representation capability.

To transition from hand-crafted network construction to the proposed graph-structured representation learning paradigm (shown in Fig. 1(b)), our framework reformulates individual brain analysis through two core technical designs: (1) a parametric, *high-order* correlation estimator that unifies linear and nonlinear dependencies, and (2) an *end-to-end* joint optimization strategy coupled with a structure-aware denoising regularizer. This design ensures that the learned topology inherently possesses downstream-task semantics, eliminating the need for iterative message-passing in predictors. Below, the mathematical notations are first introduced.

### 2.1. Notations and Problem Definition

The brain is divided into $N$ ROIs, *i.e.*, $\mathcal{V} = \{v_1, v_2, ..., v_N\}$. The BOLD time series of the $i$-th ROI is represented by $\mathbf{x}_i = (x_{i1}, x_{i2}, \cdot, x_{iD}) \in \mathbb{R}^{1 \times D}$, where $D$ is the length of the time series. $\mathbf{X} \in \mathbb{R}^{N \times D}$ is the collection of BOLD time series of $N$ brain regions with $\mathbf{x}_i$ as $i$-th row. Note that each $\mathbf{x}_i$ can *NOT* be considered as the representation/attribute of ROI $i$ by itself, since each $x_{ij}$ is just a BOLD signal snapshot without any specific semantic meaning. To demonstrate the value of $\mathbf{x}_i$'s, their correlations should be measured.

Brain network, which models the connectivity between ROIs based on these correlations, can be denoted as a graph $G = (\mathcal{V}, \mathcal{E})$, where $\mathcal{E}$ denotes the collection of edges between nodes/ROIs. The graph topology is represented as the adjacency matrix $\mathbf{A} = [a_{ij}] \in \mathbb{R}^{N \times N}$, where $a_{ij}$ is the weight between nodes $v_i$ and $v_j$.

**Problem Definition:** For brain analysis tasks, a set of $L$ subjects' brain data $\mathcal{X} = \{\mathbf{X}^{(1)} \ldots \mathbf{X}^{(L)}\}$ and the corresponding labels $\mathcal{Y} = \{y^{(1)} \ldots y^{(L)}\}$, which indicate biological sex, presence of a disease or other properties of the brain subject, are provided. Brain network modeling aims to learn from given data $\mathcal{X}$ and $\mathcal{Y}$ by designing a function $y = f(h(\mathbf{X}))$, which is composed of a brain construction function $G = h(\mathbf{X})$ and a prediction function $y = f(G)$ based on the constructed brain network $G$. Existing brain network modeling methods employ a fixed brain construction function $h(\cdot)$ and train a prediction function $f(\cdot)$, while the proposed graph-structured brain representation framework jointly learns $h(\cdot)$ and $f(\cdot)$ in an end-to-end manner.

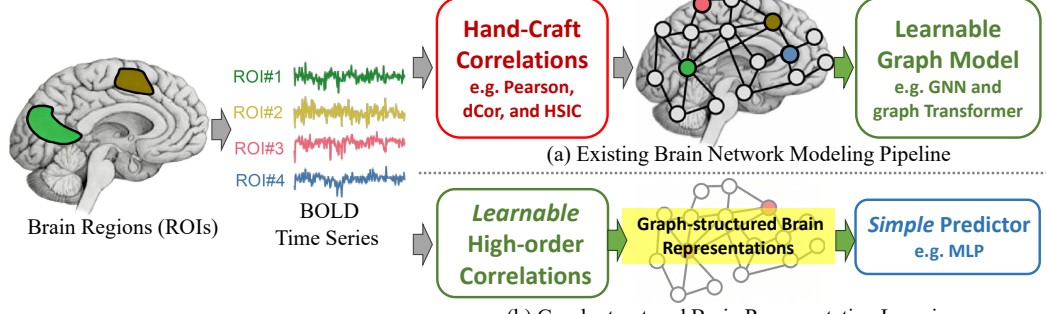

*Figure 1.* Comparison between the proposed graph-structured brain representation learning framework and the existing brain network modeling pipeline. They take the BOLD time series as input. (a) Existing modeling pipeline constructs a brain network with hand-crafted correlation coefficients without any learnable component. Then, the constructed brain network is fed into learnable flexible graph models. (b) The proposed brain representation learning framework considers the brain network as a flexible graph-structured brain representation. The representation is obtained via a learnable, high-order, parametric correlation estimator, making the downstream predictor simple.

## 2.2. From Linear/Nonlinear Correlations to High-order Dependence Measures

This section first reviews representative linear and nonlinear correlations. Then, high-order dependence measures with learnable parameters are derived by unifying them. Note that only the correlations between a pair of variables/ROIs are considered in this paper, and high-order dependence means that multiple pairs of *samples* from this pair of variables are employed. It is significantly different from the high-order correlations between multiple variables/ROIs.

### 2.2.1. LINEAR CORRELATION: PEARSON CORRELATION

Pearson Correlation Coefficient is a widely used linear correlation measure, and is the default in brain network construction. The Pearson Correlation Coefficient between $\mathbf{x}_i$ and $\mathbf{x}_j$ is defined as:

$$r_{ij} = \frac{\mathrm{Cov}(\mathbf{x}_i, \mathbf{x}_j)}{\sigma_{\mathbf{x}_i} \sigma_{\mathbf{x}_j}} = \frac{\sum_t (x_{it} - \bar{x}_i)(x_{jt} - \bar{x}_j)}{\sqrt{\sum_t (x_{it} - \bar{x}_i)^2 \sum_t (x_{jt} - \bar{x}_j)^2}},$$

where $\bar{x}_i$ and $\sigma_{\mathbf{x}_i}$ are the mean and standard deviation of $\mathbf{x}_i$, respectively. If elements in $\mathbf{x}_i$ are normalized as $\tilde{x}_{it} = (x_{it} - \bar{x}_i)/\sigma_{\mathbf{x}_i}$, $r_{ij}$ can be simplified as

$$r_{ij}^{[1]} = \sum_t \tilde{x}_{it} \tilde{x}_{jt} = \tilde{\mathbf{x}}_i \tilde{\mathbf{x}}_j^\top = (\tilde{\mathbf{x}}_i \mathbf{I}) (\tilde{\mathbf{x}}_j \mathbf{I})^\top, \quad (1)$$

where row vector $\tilde{\mathbf{x}}_i = (\tilde{x}_{i1}, \tilde{x}_{i2}, \cdot, \tilde{x}_{iD})$ and $\mathbf{I}$ is the identity matrix. As shown in Fig. 2(b), it estimates the correlation between two samples in each index, *e.g.* $x_{it}$ and $x_{jt}$ with index $t$ in Eq. (1). Although the Pearson Correlation Coefficient is computationally efficient, the characteristic of capturing linear correlation limits its capability.

### 2.2.2. NONLINEAR CORRELATION: DISTANCE CORRELATION AND HSIC

To overcome the limitation of the linear one, the nonlinear correlations are proposed to capture nonlinear dependences.

Instead of using two samples in each index in linear correlations, it employs two distances between samples with the same index pair, *e.g.* $dis(x_{it}, x_{is})$ and $dis(x_{jt}, x_{js})$ with index pair $(t, s)$ as shown in Fig. 2(c). If two random variables are independent, the covariance between pairs of sample distances approaches zero. Two representative instances of nonlinear correlation are distance Correlation (dCor) and Hilbert-Schmidt Independence Criterion (HSIC), which share a similar formula as

$$r_{ij} = \frac{1}{D^2} \mathrm{tr}(\mathbf{A}\mathbf{B}) = \frac{1}{D^2} \sum_{t,s} A_{ts} B_{ts}, \quad (2)$$

where $\mathbf{A} = (A_{ts})_{D \times D}$ and $\mathbf{B} = (B_{ts})_{D \times D}$ are the distance matrices with $A_{ts}$ and $B_{ts}$ denoting the distances of sample pairs, *i.e.*, $dis(x_{it}, x_{is})$ and $dis(x_{jt}, x_{js})$, respectively. $\mathrm{tr}(\cdot)$ represents the trace of the matrix. dCor defines the distance in Euclidean space, *i.e.*, $dis(x, y) = |x - y|$, while HSIC employs the kernel trick and defines the distance in RKHS with the Gram matrix (Schölkopf & Smola, 2002), *i.e.*, $dis(x, y) = \mathrm{kernel}(x, y)$. For simplicity, Eq. (2) can be rewritten as $r_{ij} = \frac{1}{D^2} \sum_{t,s} |x_{it} - x_{is}| \cdot |x_{jt} - x_{js}|$. By ignoring the absolute value constraint and normalization $1/D^2$, it can be formulated as

$$r_{ij}^{[2]} = \sum_{t,s} (x_{it} - x_{is}) \cdot (x_{jt} - x_{js})$$

$$= (\mathbf{x}_i \mathbf{J}) (\mathbf{x}_j \mathbf{J})^\top = \mathbf{x}_i (\mathbf{J}\mathbf{J}^T) \mathbf{x}_j^\top, \quad (3)$$

where $\mathbf{J}$ is to compute the difference of a pair of elements. $\mathbf{J} \in \mathbb{R}^{D \times D^2}$ contains $D^2$ columns, each of which corresponds to one index pair $(t, s)$ and has the following form

$$\mathbf{j}_{(t,s)} = [0, \cdots, 0, \underset{t}{1}, 0, \cdots, 0, \underset{s}{-1}, 0, \cdots, 0]^\top, \quad (4)$$

where $t$-th and $s$-th elements are $1$ and $-1$, respectively, while all other elements are $0$. dCor and HSIC compromise between the efficiency and capability of capturing nonlinear correlation. Therefore, nonlinear correlation possesses limited capability in capturing high-order dependencies.

(a) BOLD Time Series    (b) Pearson Cor.    (c) dCor. and HSIC    (d) High-order Correlations

(e) The process of high-order dependence measures approximation

*Figure 2.* Comparison between different correlation estimators. (a) The BOLD time series of two brain regions (ROIs). The representative samples are indexed with $t$, $s$, and $q$. (b) Pearson Correlation estimates linear correlation between two samples in each index, respectively. (c) dCor and HSIC estimate nonlinear correlations between relationships (distance in dCor and kernel in HSIC) in pairs of indexes. Links between samples with different colors represent different index pairs. (d) High-order Dependence measure extends to the combination of any subsets of indices. (e) An example of a high-order dependence measure.

### 2.2.3. HIGH-ORDER DEPENDENCE MEASURES

To alleviate the limitation in nonlinear correlation, high-order dependence should employ two sample tuples indexed by $t_1, t_2, \cdots, t_M$ as shown in Fig. 2(d) (Hoeffding, 1992; Lee, 2019; Sejdinovic et al., 2013). Recall that high-order dependence means that multiple pairs of *samples* from a pair of variables are employed. Motivated by U/V-statistics, its formula can be expressed as

$$r_{ij} = \frac{1}{\binom{D}{M}} \sum_{t_1, t_2, \cdots, t_M} f\Big( (x_{it_1}, x_{it_2}, \cdots, x_{it_M}),$$
$$(x_{jt_1}, x_{jt_2}, \cdots, x_{jt_M}) \Big), \quad (5)$$

where $f(\cdot, \cdot)$ denotes the product of distance, kernel, or other measures. Tensor-HSIC implements $f(\cdot, \cdot)$ with a tensor product kernel, which causes high complexity ($\mathcal{O}(D^3)$).

Note that linear correlation in Eq. (1) and approximate nonlinear correlation in Eq. (3) have a similar form. In Eq. (1), the identity matrix $\mathbf{I}$ has $D$ columns, each of which possesses one nonzero element. In Eq. (3), the difference matrix $\mathbf{J}$ has $D^2$ columns, each of which possesses two nonzero elements. Thus, Eq. (3) can also be used to approximate correlation with $M$ samples by extending $\mathbf{J}$ with $D^M$ columns and letting each column have $M$ nonzero elements, which corresponds to the index tuple $t_1, t_2, \cdots, t_M$. Unfortunately, it also leads to high computational complexity.

However, the rank of $\mathbf{J}$ in Eq. (3) is at most $D$, no matter how many columns it has. Thus, the rank of the product $\mathbf{JJ}^\top$ is at most $D$. Therefore, instead of seeking $\mathbf{J}$, the matrix $\mathbf{JJ}^\top$ can be approximated by the product of a square matrix and its transpose, *i.e.*, $\mathbf{JJ}^\top = \mathbf{OO}^\top$ where $\mathbf{O} \in \mathbb{R}^{D \times D}$, and Eq. (3) can be reformulated to

$$r_{ij}^{[high]} = \mathbf{x}_i \left( \mathbf{OO}^\top \right) \mathbf{x}_j^\top = (\mathbf{x}_i \mathbf{O})(\mathbf{x}_j \mathbf{O})^\top, \quad (6)$$

where $\mathbf{O} \in \mathbb{R}^{D \times D}$ contains the parameters of this high-order dependence estimator. Recall that each $\mathbf{x}_i$ is *NOT* the representation/attribute of ROI $i$ by itself, since each $x_{ij}$ is just a signal snapshot without any specific meaning. Thus, $\mathbf{x}_i \mathbf{O}$ should *NOT* be seen as learning an MLP for ROI $i$.

Correlation parameter $\mathbf{O}$ as well as the parameters of the predictor can be jointly trained in an end-to-end manner, as shown in Fig. 1(b). Note that the requirement of the number of columns of $\mathbf{O}$ being $D$ is critical for performance. On one hand, if the number of columns of $\mathbf{O}$ is less than $D$, it may be underfitting to $\mathbf{JJ}^\top$ with rank $D$. On the other hand, if the number of columns of $\mathbf{O}$ is much larger than $D$, it may be overfitting due to the large number of parameters. Besides, Eq. (6) can also be used to approximate the combination of multiple correlations with different orders as in Fig. 2(d).

### 2.3. End-to-end Implementation

Based on the high-order dependence estimator in Eq. (6), a scalable implementation is presented. The graph-structured brain representation $\mathbf{A}$ is obtained as follows:

$$\mathbf{A} = \sigma\Big( \text{Norm}\big(\text{TopK}(\mathbf{ZZ}^\top)\big)\Big), \quad \mathbf{Z} = \hat{\mathbf{X}} \mathbf{O}, \quad (7)$$

where $\text{TopK}(\cdot)$ selects top $K$ elements in each row, $\text{Norm}(\cdot)$ is row-wise normalization, $\sigma(\cdot)$ denotes nonlinear activation function, $\hat{\mathbf{X}}$ is the row-wise z-score normalized version of $\mathbf{X}$, and $\mathbf{X}, \hat{\mathbf{X}}, \mathbf{Z} \in \mathbb{R}^{N \times D}, \mathbf{A} \in \mathbb{R}^{N \times N}, \mathbf{O} \in \mathbb{R}^{D \times D}$. This component is designated as the **Mapping of High-order Dependence Measure (HDM)**. With the flexible graph-structured brain representation $\mathbf{A}$ in Eq. (7), the following predictor can be simple. Here, a multi-layer perceptron (MLP) and a pooling function are employed, *i.e.*,

$$\hat{y} = f_{MLP}\Big( \text{pooling}\big(f_{MLP}(\mathbf{A})\big)\Big).$$

The parameters of correlation and $f_{MLP}$'s can be jointly trained with the supervision of the cross-entropy, *i.e.*,

$\mathcal{L}_{CE} = \sum_{l=1}^{L} \text{cross} - \text{entropy}(\hat{y}^{(l)}, y^{(l)})$. To stabilize the training process, a regularization term, derived from a denoising autoencoder (DAE), is introduced to the objective function by denoising the corrupted BOLD time series with the learned brain representation $\mathbf{A}$, *i.e.*,

$$\mathcal{L} = \mathcal{L}_{CE} + \lambda \mathcal{L}_{reg}, \quad \mathcal{L}_{reg} = \left\| \mathbf{X} - \text{GNN}(\tilde{\mathbf{X}}, \mathbf{A}) \right\|,$$

where $\lambda$ is the hyper-parameter, $\tilde{\mathbf{X}}$ is the corrupted version of $\mathbf{X}$, and $\text{GNN}(\cdot, \cdot)$ is GCN to denoise the corrupted $\tilde{\mathbf{X}}$ using the learned brain network $\mathbf{A}$.

To better understand what the high-order dependence measure is, an example of a third-order dependency measure is shown in Fig. 2 (e). Three parameters $t, s, q$ in HDM can be combined through matrix product to form matrix $\mathbf{A}$, within which all pairwise interactions among the three component matrices are implicitly encoded. The matrix $\mathbf{A}$, containing all pairwise relationships among $t, s, q$, can be fed into $MLP$, which is capable of composing these interactions into an arbitrary third-order dependence measure.

## 2.4. Theoretical Analysis

This section provides a theoretical analysis to show that the proposed bilinear function in Eq. (6) can approximate any U/V-statistics in Eq. (5). To this end, the high-order U/V-statistics can be regarded as a continuous function. Specifically, $m$-order U/V-statistics can be defined as $h : K \to \mathbb{R}$ where $K \subset \mathbb{R}^{2m}$ be a compact set. Denoting by $\mathcal{H}$ the class of feedforward neural networks (MLPs) with a non-polynomial activation (*e.g.*, ReLU, sigmoid or tanh), the universal approximation theorem (UAT) (Hornik et al., 1989; Sonoda & Murata, 2017) shows that for every compact set $C$ and continuous $f : C \to \mathbb{R}$ there exists a network $g \in \mathcal{H}$ approximating $f$ uniformly on $C$ to arbitrary precision. The theoretical analysis is explored via the following three steps, and the proofs are provided in Appendix I.

- Explicitly forming bilinear features (outer-product entries) after a global invertible linear mixing and feeding them into an expressive MLP yields a class of architectures that is universal for continuous high-order dependence mappings on compact domains.
- Reducing the relation between two sample groups to a single scalar inner-product and applying a single-variable nonlinearity is strictly limited in expressivity and cannot approximate arbitrary continuous high-order dependence functions.
- Using multiple inner-product channels (multi-head inner-products) can reconstruct the full outer-product when the number of channels reaches $m^2$, and the projections are chosen appropriately; hence, multi-head constructions can recover universality.

The following theorem provides the universality of the bilinear function with the outer-product.

**Theorem 2.1.** *Let $K \subset \mathbb{R}^{2m}$ be compact and $h \in C(K)$. Let $\mathbf{W} \in \mathbb{R}^{m \times m}$ be invertible and define $T : K \to \mathbb{R}^{2m+m^2}$ as*

$$T(\mathbf{x}, \mathbf{y}) = (\mathbf{u}, \mathbf{v}, \text{vec}(\mathbf{u}\mathbf{v}^\top)), \quad \mathbf{u} = \mathbf{W}\mathbf{x}, \quad \mathbf{v} = \mathbf{W}\mathbf{y}.$$

*Then for every $\varepsilon > 0$ there exists a neural network $g \in \mathcal{H}$ (with input dimension $2m + m^2$) such that*

$$\sup_{(\mathbf{x}, \mathbf{y}) \in K} \left| h(\mathbf{x}, \mathbf{y}) - g(T(\mathbf{x}, \mathbf{y})) \right| < \varepsilon.$$

*In other words, the class $\{(\mathbf{x}, \mathbf{y}) \mapsto g(T(\mathbf{x}, \mathbf{y})) : g \in \mathcal{H}\}$ is dense in $C(K)$.*

*Remark 2.2.* The invertibility of $\mathbf{W}$ is a sufficient condition that guarantees injectivity of the intermediate linear mixing. It ensures no information about $(\mathbf{x}, \mathbf{y})$ is lost before forming bilinear features. If $\mathbf{W}$ is rank-deficient, the mapping $T$ may collapse distinct $(\mathbf{x}, \mathbf{y})$ to the same $T(\mathbf{x}, \mathbf{y})$. Then only those target functions constant on fibers of $T$ can be represented as $g \circ T$.

However, the inner-product bilinear function, *i.e.*, bilinear function without the outer-product, can not universally approximate continuous functions. To this end, the single-scalar pipeline function class is defined as follows, and theoretical analysis is given in Theorem 2.4.

**Definition 2.3.** The single-scalar pipeline function class is

$$\mathcal{F}_1 := \{(\mathbf{x}, \mathbf{y}) \mapsto \phi(\mathbf{x}^\top M \mathbf{y}) : M \in \mathbb{R}^{m \times m}, \ \phi \in C(\mathbb{R})\}.$$

**Theorem 2.4.** *Let $m \geq 2$ and let $K \subset \mathbb{R}^{2m}$ be any compact set that contains a set of the form $\{(\mathbf{x}, \mathbf{y}_0) : \mathbf{x} \in U\}$ where $U \subset \mathbb{R}^m$ contains two points $\mathbf{x}^{(1)} \neq \mathbf{x}^{(2)}$ with $\mathbf{x}^{(1)} - \mathbf{x}^{(2)} \notin \ker(M\mathbf{y}_0)$ for every $M$ in a prescribed collection (in particular one may take $U$ to have nonempty interior). Then there exists a continuous function $h \in C(K)$ and a constant $\delta > 0$ such that for every $f \in \mathcal{F}_1$,*

$$\|h - f\|_{L^\infty(K)} \geq \delta.$$

*Consequently, $\mathcal{F}_1$ is not dense in $C(K)$.*

*Remark 2.5.* The intuitive reason is that for fixed $\mathbf{y}$, each $f \in \mathcal{F}_1$ reduces to a single-variable function of a one-dimensional linear projection of $\mathbf{x}$, while generic continuous functions on $\mathbf{x}$ cannot be represented through a single linear projection and a scalar nonlinearity.

To alleviate the limitation of the inner-product bilinear function on universal approximation, *multi-head* inner-products are proposed to approximate the outer-product. The following theorem provides the approximation error.

**Theorem 2.6.** *Let $m \geq 1$ and let $K \subset \mathbb{R}^{2m}$ be compact. For $R = m^2$ define matrices $\{\mathbf{U}^{(i,j)}, \mathbf{V}^{(i,j)}\}_{1 \leq i,j \leq m}$ by*

$$\mathbf{U}^{(i,j)} := \mathbf{e}_i^\top \in \mathbb{R}^{1 \times m}, \qquad \mathbf{V}^{(i,j)} := \mathbf{e}_j^\top \in \mathbb{R}^{1 \times m},$$

where $\mathbf{e}_k$ is the $k$-th standard basis column vector in $\mathbb{R}^m$. Define scalar channels

$$s_{ij}(\mathbf{x}, \mathbf{y}) := (\mathbf{U}^{(i,j)}\mathbf{x})^\top (\mathbf{V}^{(i,j)}\mathbf{y}) = \mathbf{x}_i \mathbf{y}_j.$$

Let $s(\mathbf{x}, \mathbf{y}) \in \mathbb{R}^{m^2}$ be the vector obtained by ordering $\{s_{ij}\}_{i,j}$. Then $s(\mathbf{x}, \mathbf{y}) = \mathrm{vec}(\mathbf{xy}^\top)$ and the mapping $(\mathbf{x}, \mathbf{y}) \mapsto s(\mathbf{x}, \mathbf{y})$ is continuous and injective on any set where $(\mathbf{x}, \mathbf{y}) \mapsto \mathbf{xy}^\top$ is injective. Consequently, for any continuous $h : K \to \mathbb{R}$ and any $\varepsilon > 0$ there exists an MLP $g \in \mathcal{H}$ such that

$$\sup_{(\mathbf{x}, \mathbf{y}) \in K} \left| h(\mathbf{x}, \mathbf{y}) - g(s(\mathbf{x}, \mathbf{y})) \right| < \varepsilon.$$

*Remark* 2.7. The matrices chosen in the constructive proof are the simplest possible: each channel extracts exactly one pairwise product $\mathbf{x}_i \mathbf{y}_j$. In practice one often uses fewer channels $R < m^2$ with learned or random projection matrices $\mathbf{U}^{(r)}, \mathbf{V}^{(r)}$ and relies on the downstream MLP to recover or approximate the necessary combinations.

*Remark* 2.8. By combining Theorems 2.1 and 2.6, the bilinear function with multi-head inner-products possesses the property of universal approximation.

# 3. Experiment

This section provides a comprehensive evaluation of the proposed BRep with the setup described in Appendix B.

## 3.1. Experimental Results

**Brain Disorder Classification.** Tab. 1 reports the performance comparison between the proposed BRep and representative baselines on the ABIDE and ADHD-200. Overall, the proposed BRep consistently achieves competitive performance, outperforming most baselines in terms of AUC, ACC, SEN, and SPE. The performance of BRep depends on the quality of the input brain network, not only on advanced architectural design. Although GNNs and GTs-based models have attained promising performance by relying on sophisticated architectural designs, the proposed BRep, based on learnable brain network construction and simple NNs, achieves superior results. For example, on the ADHD-200 dataset, the proposed BRep shows the best results on nearly all metrics. Particularly, the models outperform the second-best BQN by 2.14% on ACC.

## 3.2. Interpretability Analysis

**Case Study.** This experiment aims to assess the biological interpretability of the proposed BRep by visualizing differential brain networks, *i.e.*, connectivity differences between Normal Controls (NC) and Autism Spectrum Disorder (ASD) groups. In the first step, group-level connectivity templates are calculated by averaging the functional connectivity matrices within each group: $A_{\text{Template}}^{\text{ASD}} = \frac{1}{n_1} \sum_{i=1}^{n_1} A_i^{\text{ASD}}$,

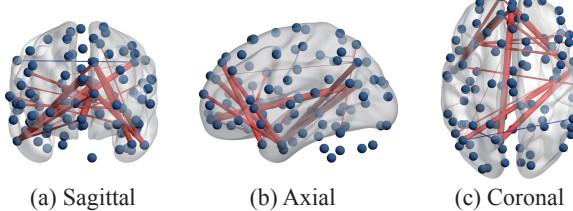

| (a) Sagittal | (b) Axial | (c) Coronal |

*Figure 3.* Top-20 differential connectivity, shown in 3 views.

$A_{\text{Template}}^{\text{NC}} = \frac{1}{n_2} \sum_{i=1}^{n_2} A_i^{\text{NC}}$, where $n_1$ denotes the number of ASD patients, and $n_2$ denotes the number of normal subjects. Then, the connectivity difference matrix can be obtained by $A_{\text{Template}}^{\text{ASD}} - A_{\text{Template}}^{\text{NC}}$. For a comprehensive description of the time series preprocessing pipeline, refer to Appendix C.2. Finally, the edges with the 20 largest absolute values are visualized in Figs. 3 and 9. Figs. 4 and 5 show the BRep-learned and VGAE-learned (Kipf & Welling, 2016) NC and ASD brain connectivity matrices, along with their difference (ASD − NC), on ABIDE. To clearly demonstrate their biological interpretability, all ROIs are reordered into six atlas-based macro–anatomical groups – Prefrontal (PFR), Frontal (FR), Parietal (PR), Occipital (OR), Temporal (TR), and Subcortical (SUB).

At the macro–anatomical level, the NC and ASD templates (Fig. 4 (a) and (b)) show a recognizable modular structure: within-block connectivity (block diagonals) is stronger than between-block connectivity (Power et al., 2011). The NC and ASD brain connections remain visually similar, which is consistent with ASD being a neurodevelopmental condition with subtle, distributed FC alterations rather than gross disruption of whole networks (Holiga et al., 2019). Note that the difference of BRep-learned connections (Fig. 4 (c)) shows spatially coherent clusters of altered connectivity, mainly in TR–FR/PFR, TR–OR, TR–PR, and SUB–PFR connections, corresponding to atypical coupling between sensory, visual, social–cognitive, and subcortical–prefrontal systems that have frequently been reported in ASD studies (Supekar et al., 2013; Long et al., 2016; Uddin et al., 2013b; Woodward et al., 2017). These patterns are interpretable from a neuroscience perspective.

From the ROI level connectivity difference perspective, the differential connectivity analysis reveals widespread *hyperconnectivity* in ASD relative to NC (Supekar et al., 2013), indicating stronger functional connectivity in ASD. Specifically, enhanced connectivity is observed in the anterior cingulate cortex (ACC) and bilateral insula, which are critical regions of the salience network (Uddin et al., 2013a). Additional hyperconnectivity is detected in the frontotemporal regions, particularly in the posterior superior temporal sulcus (pSTS), which is closely associated with biological motion processing, social cognition, and speech perception (Uddin et al., 2013a), as well as in the middle temporal gyrus (MTG), motor, and visual systems. Moreover, cross-

*Table 1.* Performance comparison with three types of baselines, reported as percentages ($mean_{\pm std}$) over five trials. **Bold** and underlined indicate the best and second-best results, respectively.

| Type | Model | ABIDE | | | | ADHD-200 | | | |
|------|-------|-------|-------|-------|-------|----------|-------|-------|-------|
| | | AUC ↑ | ACC ↑ | SEN ↑ | SPE ↑ | AUC ↑ | ACC ↑ | SEN ↑ | SPE ↑ |
| *GNNs* | GCN | $59.59_{\pm3.44}$ | $59.30_{\pm3.38}$ | $56.67_{\pm4.37}$ | $61.55_{\pm5.29}$ | $67.01_{\pm3.56}$ | $64.92_{\pm6.32}$ | $65.09_{\pm6.03}$ | $62.24_{\pm5.90}$ |
| | GAT | $60.43_{\pm3.88}$ | $60.10_{\pm4.13}$ | $59.26_{\pm5.51}$ | $62.89_{\pm8.03}$ | $61.97_{\pm3.28}$ | $63.38_{\pm3.18}$ | $64.54_{\pm13.97}$ | $45.10_{\pm18.34}$ |
| | BrainGNN | $64.42_{\pm3.57}$ | $63.09_{\pm1.35}$ | $65.65_{\pm2.88}$ | $60.67_{\pm3.68}$ | $67.19_{\pm2.86}$ | $65.16_{\pm3.81}$ | $65.09_{\pm2.11}$ | $64.43_{\pm3.81}$ |
| | BrainGB | $70.32_{\pm3.66}$ | $65.12_{\pm3.90}$ | $67.01_{\pm10.00}$ | $60.07_{\pm8.53}$ | $75.23_{\pm11.02}$ | $69.34_{\pm7.41}$ | $67.46_{\pm9.82}$ | $68.15_{\pm8.41}$ |
| | FBNetGen | $74.55_{\pm3.77}$ | $67.09_{\pm3.37}$ | $64.71_{\pm9.85}$ | $69.61_{\pm9.30}$ | $77.40_{\pm4.76}$ | $68.82_{\pm6.27}$ | $66.45_{\pm7.73}$ | $71.56_{\pm14.07}$ |
| | A-GCL | $73.86_{\pm2.91}$ | $71.04_{\pm2.40}$ | $71.42_{\pm3.03}$ | $70.95_{\pm3.19}$ | $74.78_{\pm4.39}$ | $73.11_{\pm4.30}$ | $72.04_{\pm4.68}$ | $73.08_{\pm4.10}$ |
| *GTs* | SAN | $71.35_{\pm2.18}$ | $65.34_{\pm2.91}$ | $55.41_{\pm9.29}$ | $68.39_{\pm7.50}$ | $51.22_{\pm2.21}$ | $51.09_{\pm2.00}$ | $50.43_{\pm19.32}$ | $51.74_{\pm20.16}$ |
| | Graphormer | $63.91_{\pm4.05}$ | $61.88_{\pm6.85}$ | $66.30_{\pm9.98}$ | $55.74_{\pm11.00}$ | $58.64_{\pm1.50}$ | $61.60_{\pm0.90}$ | $73.34_{\pm2.90}$ | $33.96_{\pm6.10}$ |
| | GraphTrans | $60.13_{\pm6.73}$ | $57.83_{\pm4.71}$ | $65.70_{\pm10.30}$ | $49.77_{\pm11.52}$ | $51.49_{\pm1.15}$ | $50.76_{\pm2.07}$ | $62.39_{\pm9.43}$ | $39.13_{\pm10.74}$ |
| | BrainNETTF | $77.93_{\pm1.41}$ | $69.26_{\pm2.26}$ | $65.92_{\pm8.60}$ | **$73.20_{\pm6.06}$** | $79.79_{\pm3.14}$ | $72.67_{\pm3.17}$ | $73.64_{\pm11.06}$ | $72.08_{\pm5.66}$ |
| | ContrastPool | $57.36_{\pm0.87}$ | $57.44_{\pm0.69}$ | $57.66_{\pm6.85}$ | $57.08_{\pm7.79}$ | $71.19_{\pm2.26}$ | $69.16_{\pm2.85}$ | $67.71_{\pm3.15}$ | $70.59_{\pm2.91}$ |
| | ALTER | $77.99_{\pm2.21}$ | $70.10_{\pm2.26}$ | $72.84_{\pm7.40}$ | $67.68_{\pm5.81}$ | $83.16_{\pm1.61}$ | $73.48_{\pm1.38}$ | $74.58_{\pm6.85}$ | $72.20_{\pm5.82}$ |
| | BioBGT | $69.96_{\pm1.18}$ | $69.70_{\pm2.92}$ | $67.04_{\pm3.41}$ | $72.02_{\pm4.67}$ | $71.64_{\pm1.14}$ | $71.06_{\pm0.08}$ | $75.39_{\pm5.45}$ | $71.92_{\pm2.29}$ |
| *NNs* | MLP | $75.60_{\pm2.38}$ | $70.92_{\pm2.34}$ | $63.96_{\pm9.58}$ | $73.03_{\pm7.68}$ | $78.36_{\pm1.88}$ | $70.68_{\pm4.98}$ | $74.15_{\pm4.63}$ | $64.42_{\pm7.60}$ |
| | BQN | **$79.85_{\pm1.27}$** | $72.53_{\pm1.41}$ | $73.26_{\pm5.99}$ | $72.03_{\pm6.24}$ | $83.34_{\pm1.13}$ | $75.68_{\pm1.95}$ | **$79.73_{\pm2.27}$** | $71.63_{\pm4.87}$ |
| *Ours* | BRep | $77.64_{\pm2.06}$ | **$73.58_{\pm3.67}$** | **$74.86_{\pm8.16}$** | $68.12_{\pm5.21}$ | **$84.53_{\pm2.85}$** | **$77.82_{\pm3.50}$** | $78.11_{\pm3.12}$ | **$73.30_{\pm4.90}$** |

*Table 2.* Comparison of baseline models and their +HDM variants on ABIDE and ADHD-200. Best models are highlighted in red.

| Model | ABIDE | | | | ADHD-200 | | | |
|-------|-------|-------|-------|-------|----------|-------|-------|-------|
| | AUC ↑ | ACC ↑ | SEN ↑ | SPE ↑ | AUC ↑ | ACC ↑ | SEN ↑ | SPE ↑ |
| GCN | $59.59_{\pm3.44}$ | $59.30_{\pm3.38}$ | $56.67_{\pm4.37}$ | $61.55_{\pm5.29}$ | $67.01_{\pm3.56}$ | $64.92_{\pm6.32}$ | $65.09_{\pm6.03}$ | $62.24_{\pm5.90}$ |
| GCN+HDM | $62.06_{\pm3.95}$ | $63.34_{\pm8.78}$ | $64.50_{\pm9.10}$ | $62.20_{\pm8.40}$ | $73.24_{\pm0.83}$ | $70.02_{\pm2.61}$ | $75.64_{\pm0.98}$ | $64.07_{\pm1.68}$ |
| BrainNETTF | $77.93_{\pm1.41}$ | $69.26_{\pm2.26}$ | $65.92_{\pm8.60}$ | $73.20_{\pm6.06}$ | $79.79_{\pm3.14}$ | $72.67_{\pm3.17}$ | $73.64_{\pm11.06}$ | $72.08_{\pm5.66}$ |
| BrainNETTF+HDM | $77.61_{\pm2.45}$ | $71.61_{\pm3.25}$ | $68.64_{\pm7.80}$ | $74.08_{\pm4.84}$ | $83.04_{\pm2.01}$ | $76.19_{\pm2.08}$ | $75.28_{\pm10.84}$ | $75.38_{\pm7.67}$ |
| BQN | $79.85_{\pm1.27}$ | $72.53_{\pm1.41}$ | $73.26_{\pm5.99}$ | $72.03_{\pm6.24}$ | $83.34_{\pm1.13}$ | $75.68_{\pm1.95}$ | $79.73_{\pm2.27}$ | $71.63_{\pm4.87}$ |
| BQN+HDM | $77.59_{\pm1.85}$ | $73.15_{\pm1.37}$ | $73.95_{\pm4.15}$ | $72.56_{\pm4.68}$ | $83.44_{\pm1.01}$ | $79.97_{\pm3.34}$ | $76.97_{\pm5.81}$ | $73.25_{\pm9.84}$ |

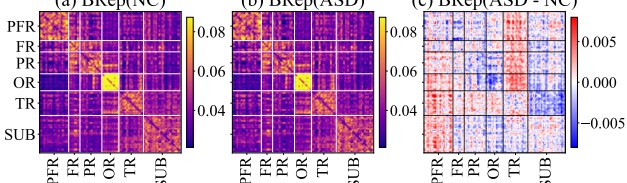

*Figure 4.* BRep-learned NC, ASD, and their difference.

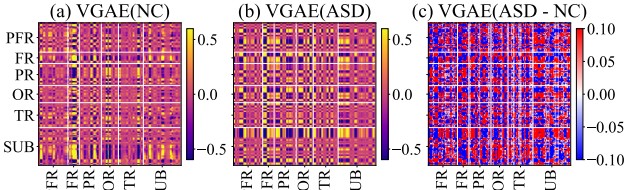

*Figure 5.* VGAE-learned NC, ASD, and their difference.

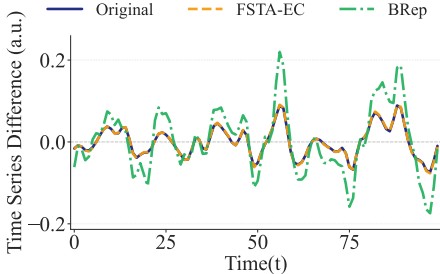

*Figure 6.* Time series differences: Original, FSTA-EC, BRep.

modal integration areas such as the occipito-temporal cortex (OT) and posterior middle temporal gyrus (pMTG) also exhibit increased connectivity (Hong et al., 2019). These ROI level findings are consistent with the spatially coherent patterns observed in Fig. 4(c). At the subsystem level, ASD patients demonstrate higher within-system connectivity in primary sensory, paralimbic, and association networks, along with stronger between-system connectivity

across sensory–paralimbic, sensory–association, and paralimbic–association systems (Supekar et al., 2013). By aligning with prior reports of atypical functional connectivity in ASD, these results are evidence for the biological interpretability of BRep. These subsystem-level alterations are also consistent with the observations in Fig. 4(c).

To further assess the reliability of the biological interpretability of BRep, Figs. 4 and 5 are compared. It is clear that VGAE-learned brain connectivity matrices do not exhibit a clear FC pattern or strong within-module cohesion; both the connectivity and the ASD – NC difference are dominated by rapid sign changes at the single-ROI level with little

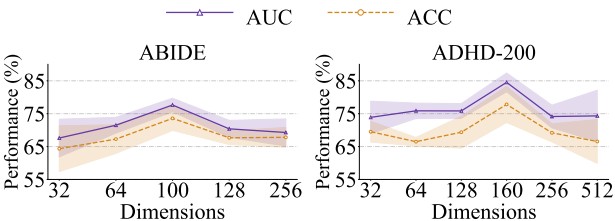

Figure 7. Impact of the dimension of the mapping of HDM.

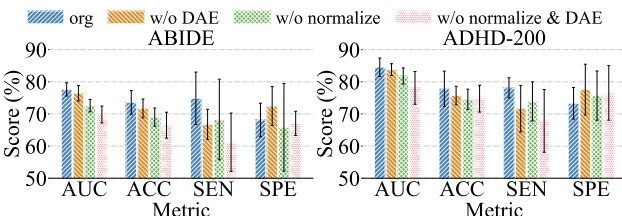

Figure 8. Impact of the denoising module and normalization.

alignment to lobar/subcortical groups. This lack of modular and spatially coherent structure makes the VGAE-learned connectivity harder to relate to known functional networks.

Taken together, these observations suggest that BRep-learned FC patterns are more structured and spatially coherent, while demonstrating consistency across macro–anatomical, subsystem, and ROI levels. Overall, this qualitative comparison indicates that BRep provides connectivity estimates that are biologically interpretable.

**Time Series Comparison.** The experiment aims to evaluate whether different models can enhance the discriminative temporal patterns between ASD and NC. To this end, the time series are first divided into ASD and NC groups and then processed by the respective models (original, FSTA-EC, and BRep). Differential time series are computed following the procedure described in the **Case Study**, and the resulting matrices are averaged over the ROI dimension to derive group-level differential curves. Detailed processing steps are provided in Appendix C.2. Fig. 6 presents the differential time series curves obtained from the three approaches. The curve produced by FSTA-EC closely resembles the raw time series, suggesting limited enhancement. In contrast, BRep yields more pronounced fluctuations and distinctive patterns, implying that it captures additional discriminative temporal dynamics beyond those present in the original signals. In summary, these findings suggest that BRep provides stronger representational capacity for distinguishing ASD from NC. Details can be found in Appendix C.3.

**Applicability Analysis.** To further extend the evaluation of HDM to advanced GNNs and GTs, Tab. 2 presents the results of comparing GCN and BrainNETTF with their +HDM counterparts on ABIDE and ADHD-200 datasets. It can be observed that there are consistent improvements across most metrics. In particular, GCN+HDM achieves an ACC gain of approximately $5\%$ and a SEN gain of over $10\%$ on ADHD-200. These improvements confirm that HDM enhances the performance of not only *NNs* but also *GNN-* and *GT*-based models, underscoring its broader effectiveness.

**Analysis of the Mapping of High-order Dependence Measure (HDM).** Fig. 7 illustrates the impact of the dimension $D$ of the HDM on model performance for ABIDE and ADHD-200. The results show that peak performance is achieved when $D = 100$ on ABIDE and $D = 160$ on

ADHD-200. This is consistent with the theoretical statement in Section 2.2.3, since the time series dimension is also $100$ and $160$, respectively, which makes the parameter matrix $\mathbf{O}$ square. Such a configuration is expected to better capture correlations. In contrast, smaller dimensions tend to reduce performance, likely due to underfitting, whereas excessively large dimensions (*e.g.*, $256$, $512$) may lead to a decline in performance, possibly due to overfitting. In summary, these confirm that aligning the dimension of the correlation estimator with that of the time series is important.

### 3.3. Ablation Study

To evaluate the contributions of the employed denoising module and normalization strategy, three ablation variants are designed by removing denoising, normalization, or both, and compare their performance with the original BRep on the ABIDE and ADHD-200 datasets (Fig. 8). The results show consistent performance degradation when either component is removed. In particular, eliminating both normalization and denoising leads to the largest drop, with ACC reduced by about $7\%$ and AUC by $7.84\%$ compared with the full model. Moreover, removing only one component (either denoising or normalization) still causes noticeable decreases across multiple metrics, especially in SEN and SPE. These results indicate their importance in enhancing discriminative power and stability.

## 4. Conclusions

This work has introduced the Brain Representation (BRep) learning problem by revisiting the construction of brain functional networks. Rather than relying on fixed, hand-crafted correlations of BOLD time series, the study enhances linear and nonlinear correlations into high-order, parametric, and learnable forms, further implemented by combining with a TopK sparsification strategy. In this way, the brain network serves as a flexible, graph-structured representation, enabling the predictor to remain simple while maintaining an end-to-end framework. Theoretical analysis guarantees the model's universal approximation to any U/V-statistics. Extensive evaluations have demonstrated that the proposed BRep achieves superior predictive performance, high efficiency, universal applicability, and interpretability. These indicate the promising direction of transitioning from hand-crafted correlations to learnable ones.

## Acknowledgements

This work was supported in part by the National Natural Science Foundation of China (No. 92570118, U22B2036, 62376088, 62272020, 62025604, 92370111, 62272340, 62261136549, 52441501), in part by the Hebei Natural Science Foundation (No. F2024202047), in part by the National Science Fund for Distinguished Young Scholarship (No. 62025602), in part by the Hebei Yanzhao Golden Platform Talent Gathering Programme Core Talent Project (Education Platform) (HJZD202509), in part by the Post-graduate's Innovation Fund Project of Hebei Province (CXZZBS2025036), in part by the Tencent Foundation, and in part by the XPLORER PRIZE.

## Impact Statement

This paper presents work whose goal is to advance the field of Machine Learning. There are many potential societal consequences of our work, none of which we feel must be specifically highlighted here.

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

## A. Related Works

Resting-state fMRI functional connectivity has provided a foundation for characterizing large-scale brain networks. Building on this, graph-based deep learning methods, including Graph Neural Networks (GNNs), Graph Transformers (GTs), and Neural Networks (NNs), have been applied to brain network modeling, while effective connectivity (EC) studies offer a complementary causal perspective.

**Resting-State fMRI Functional Connectivity.** Resting-state fMRI (fMRI) has become a widely used tool for mapping large-scale brain networks and individual differences (Van Den Heuvel & Pol, 2010). Since the seminal work of Biswal et al. (1995), functional connectivity (FC) has been inferred from low-frequency blood-oxygen-level-dependent (BOLD) fluctuations (0.01–0.1 Hz) (Cordes et al., 2001). Common approaches include correlation-based analysis, independent component analysis (ICA), clustering, and graph-theoretical modeling (Beckmann et al., 2005; Achard et al., 2006). fMRI FC has provided valuable insights into cognition and has been linked to various neurological and psychiatric disorders, such as Alzheimer's disease and schizophrenia (Greicius et al., 2004; Fair et al., 2009). However, existing fMRI–based FC suffers from limited representational capacity, making it difficult to capture complex spatiotemporal dependencies among brain regions (ROIs) (Friston, 2011). **Graph Neural Networks (GNNs)** have been widely applied to FC analysis in recent years, aiming to learn representations of ROIs and connectivity patterns. Representative works include BrainGNN (Li et al., 2021), BrainGB (Cui et al., 2022), FBNetGen (Kan et al., 2022a) and A-GCL (Zhang et al., 2023). These methods typically rely on local neighborhood aggregation over FC-derived graphs, which may overlook global interactions among ROIs. **Graph Transformers (GTs)** were subsequently introduced to address this limitation. By employing global self-attention, GTs are able to capture holistic inter-regional interactions and long-range dependencies. Notable examples include BrainNETTF (Kan et al., 2022b), ContrastPool (Xu et al., 2024), ALTER (Yu et al., 2024), and BioBGT (Peng et al., 2025). However, these approaches often conflate functional connectivity matrices with ROI features, where correlation coefficient matrices are simultaneously treated as both adjacency matrices and node features. More recently, **Neural Networks (NNs)** have been proposed to address this issue by treating the Pearson Correlation Coefficient matrix as a single input and employing non–message-passing mechanisms to model brain functional connectivity. For instance, BQN (Yang et al., 2025) introduces a simple yet effective Quadratic Neural Network specifically designed for modeling brain functional connectivity.

**Brain Effective Connectivity.** Early studies on effective connectivity (EC) relied on model-driven methods such as SEM (Eisenhauer et al., 2015), DCM (Friston et al., 2012; 2014), and GC (DSouza et al., 2017), which provide causal interpretability but depend heavily on prior assumptions. Subsequent approaches, including CTE-score (Liu et al., 2021) and VACOEC (Liu et al., 2019), sought to better capture temporal information and improve search efficiency. Recently, deep learning models like STGCMEC (Zou et al., 2022) and FSTA-EC (Xiong et al., 2025) have emerged, marking a shift from model-driven to data-driven paradigms in EC research.

**End-to-End Learnable Graph Construction.** Recent studies explore end-to-end graph structure learning for functional connectivity (Mahmood et al., 2021; Noman et al., 2025). However, these methods treat constructed graphs merely as intermediate topologies requiring complex downstream architectures (*e.g.*, heavy GNNs or Graph Transformers) for message passing. In contrast, BRep conceptualizes the graph itself as a rich, high-order semantic representation, simplifying the downstream predictor to a basic MLP.

## B. Experimental Setup

**Datasets.** The experiments utilize four benchmark fMRI datasets for brain network analysis:

- Autism Brain Imaging Data Exchange (**ABIDE**)[1]. It combines functional and structural brain imaging data from seventeen international sites to study the neural bases of autism. It comprises 1,009 subjects, including 516 individuals with Autism Spectrum Disorder (ASD) and 493 normal controls (NC) (Craddock et al., 2012).

- Attention Deficit Hyperactivity Disorder (**ADHD-200**)[2]. It is a multi-site dataset to study neural basis of Attention Deficit Hyperactivity Disorder (ADHD-200). The experiment utilizes 459 subjects, specifically, 230 developing individuals and 229 ADHD patients.

---

[1]http://preprocessed-connectomes-project.org/abide/
[2]https://fcon_1000.projects.nitrc.org/indi/adhd200/

- Alzheimer's Disease Neuroimaging Initiative (**ADNI**)[3]. It is a multi-center, longitudinal research program designed to systematically collect cognitive assessments, neuroimaging data, and associated biomarkers to enable quantitative analysis of patterns related to Alzheimer's disease progression. The subset used in this study comprises 538 subjects, categorized into four predefined groups with sample sizes of 54, 79, 194, and 211, respectively.

- Parkinson's Progression Markers Initiative (**PPMI**). It is a large-scale, multi-center longitudinal study aimed at identifying biological markers associated with Parkinson's disease risk, onset, and progression. The dataset encompasses four clinically distinct subject groups: 15 normal controls (NC), 14 individuals with SWEDD (scans without evidence of dopaminergic deficit), 67 prodromal subjects, and 113 patients diagnosed with Parkinson's disease (PD).

Following Craddock et al. (2012), ROIs for both datasets are defined using the Craddock 200 atlas, with 200 ROIs in ABIDE and 190 ROIs in ADHD-200. For ADNI, resting-state fMRI data were preprocessed using the Data Processing Assistant for Resting-State fMRI (DPARSF) toolkit, followed by region definition based on the AAL90 atlas. For PPMI, regions of interest were delineated using the AAL116 atlas, and all preprocessing procedures were completed by Xu et al. (2023). Following Yang et al. (2025); Kan et al. (2022b), all datasets are randomly split into training, validation, and test sets with a ratio of 7/1/2.

**Baselines.** The proposed BRep is evaluated by comparing its performance with 14 baseline models. Based on their architecture, they are grouped into the following three categories.

- Graph Neural Networks (*GNNs*), including classic GCN (Kipf, 2016) and GAT (Veličković et al., 2017), and four brain-specific GNNs, *i.e.*, BrainGNN (Li et al., 2021), BrainGB (Cui et al., 2022), FBNetGen (Kan et al., 2022a), and A-GCL (Zhang et al., 2023);

- Graph Transformers (*GTs*), including three general GTs, that is, SAN (Kreuzer et al., 2021), Graphormer (Ying et al., 2021), and GraphTrans (Wu et al., 2021), and four brain-specific GTs, *i.e.*, BrainNETTF (Kan et al., 2022b), ContrastPool (Xu et al., 2024), ALTER (Yu et al., 2024), and BioBGT (Peng et al., 2025);

- Neural Network (*NNs*), that is, MLP (Rumelhart et al., 1986) and BQN (Yang et al., 2025).

**Metrics.** The aim is to diagnose ASD on the ABIDE dataset and ADHD on the ADHD-200 dataset, both formulated as binary classification tasks. Considering the biomedical nature, the model is evaluated using four metrics: (1) Area Under the ROC Curve (AUC), quantifying the trade-off between true positive and false positive rates; (2) Accuracy (ACC), indicating the proportion of correctly classified samples; (3) Sensitivity (SEN), quantifying the model's ability to identify positive cases; and (4) Specificity (SPE), indicating the model's ability to identify negative cases.

For multi-class datasets such as ADNI and PPMI, and considering their biomedical nature, the evaluation is conducted using five metrics: (1) Area Under the ROC Curve (AUC), computed using the one-vs-rest scheme and macro-averaged across classes; (2) Accuracy (ACC), measuring the overall proportion of correctly classified samples; (3) F1 score (F1), calculated as the macro-average of per-class F1 scores; (4) Sensitivity (SEN), obtained as the macro-average of per-class recall; (5) Specificity (SPE), derived from the multi-class confusion matrix and macro-averaged across classes.

**Implementation Details.** All experiments are conducted in PyTorch on a Linux machine equipped with a single GeForce RTX 3090 24GB GPU. The models are tuned under a semi-supervised learning framework, and hyperparameters are selected via a grid search strategy. All baseline models are reproduced using the hyperparameters reported in their original papers. The Adam optimizer is used with an initial learning rate of $1 \times 10^{-4}$, a target learning rate of $1 \times 10^{-5}$, and a weight decay of $1 \times 10^{-3}$. Both cross-entropy and mean squared error (MSE) losses are utilized for training. Dropout rates are selected from the set $\{0.0, 0.1, 0.2, 0.3\}$. Final results are reported as the mean and standard deviation over five random runs.

## C. Experimental Supplement

### C.1. Circle Plots of Differential Brain Connections

Fig. 9 (a) offers a complementary circular visualization with more detailed brain-region annotations. Specifically, it depicts the top-20 differential connections, where red and blue lines denote positive and negative values, respectively. This

---

[3]https://adni.loni.usc.edu/

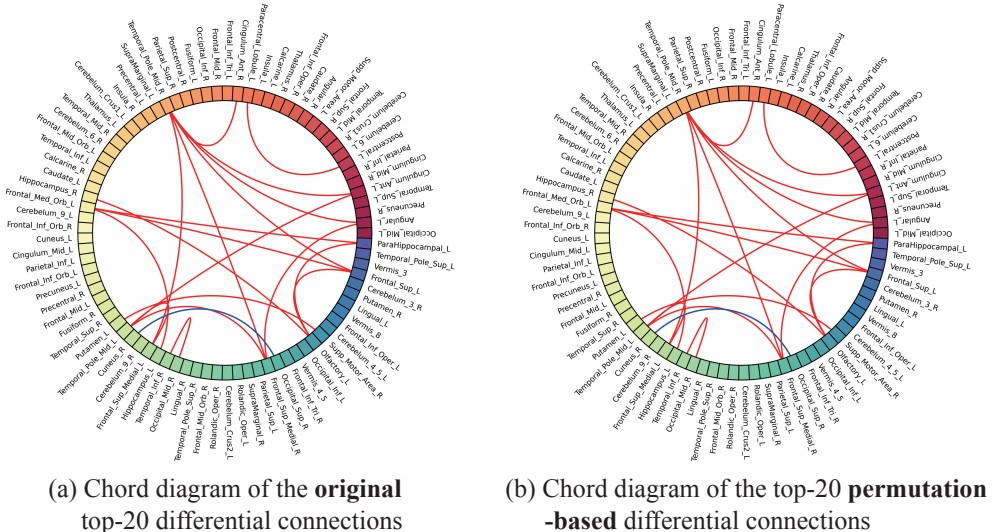

(a) Chord diagram of the **original** top-20 differential connections

(b) Chord diagram of the top-20 **permutation -based** differential connections

*Figure 9.* Circular visualization of the **original** and **permutation-based** top-20 differential connections between ASD and NC identified by BRep. Red and Blue lines represent positive and negative values, respectively.



*Figure 10.* Time Series heatmaps of NC subjects under different processing methods (Original, FSTA-EC, and BRep)



*Figure 11.* Time Series heatmaps of ASD subjects under different processing methods (Original, FSTA-EC, and BRep)

representation offers a clearer view of how the connections are distributed across specific cortical and subcortical regions, thereby enhancing the interpretability of the results.

To ensure the statistical rigor of the interpretability analysis, we conducted 2000 edge-wise permutation testing on the BRep learned matrix $A$ of all 493 NC and 516 ASD, followed by Benjamini–Hochberg FDR correction (q<0.05), as shown in Fig. 9 (b). Only connections that remained significant after correction were retained. The resulting set of significant edges shows strong overlap with those exhibiting the largest raw ASD – NC differences, indicating that the model-identified group differences are stable, reliable, and unlikely to be driven by noise. Furthermore, the perturbed top-20 difference matrix closely matches the original top-20 difference matrix, demonstrating that the observed patterns are robust to random fluctuations.

*Table 3.* Comparison of four baseline models and +HDM variants on multi-class datasets (ADNI, PPMI) ($mean_{\pm std}$). Best models are highlighted in red.

| Model | ADNI | | | | | PPMI | | | | |
|---|---|---|---|---|---|---|---|---|---|---|
| | AUC ↑ | ACC ↑ | F1 ↑ | SEN ↑ | SPE ↑ | AUC ↑ | ACC ↑ | F1 ↑ | SEN ↑ | SPE ↑ |
| GCN | $53.99_{\pm 1.51}$ | $41.67_{\pm 4.84}$ | $22.54_{\pm 1.95}$ | $36.31_{\pm 2.35}$ | $75.77_{\pm 1.41}$ | $59.58_{\pm 4.86}$ | $56.00_{\pm 3.90}$ | $23.85_{\pm 4.00}$ | $27.96_{\pm 2.44}$ | $76.89_{\pm 1.60}$ |
| GCN+HDM | $60.46_{\pm 2.19}$ | $41.92_{\pm 5.10}$ | $23.98_{\pm 3.20}$ | $32.32_{\pm 3.05}$ | $76.48_{\pm 1.88}$ | $65.31_{\pm 5.34}$ | $54.08_{\pm 7.42}$ | $27.33_{\pm 5.22}$ | $30.02_{\pm 3.54}$ | $78.38_{\pm 2.30}$ |
| BrainNETTF | $63.86_{\pm 1.84}$ | $50.40_{\pm 1.67}$ | $30.47_{\pm 3.32}$ | $34.34_{\pm 1.70}$ | $80.28_{\pm 0.64}$ | $59.08_{\pm 2.19}$ | $51.67_{\pm 6.24}$ | $25.86_{\pm 4.92}$ | $27.64_{\pm 2.70}$ | $75.07_{\pm 1.24}$ |
| BrainNETTF+HDM | $68.50_{\pm 1.88}$ | $52.70_{\pm 3.50}$ | $31.85_{\pm 2.12}$ | $36.90_{\pm 2.41}$ | $82.90_{\pm 1.43}$ | $61.03_{\pm 3.93}$ | $52.50_{\pm 5.24}$ | $32.98_{\pm 8.92}$ | $34.64_{\pm 6.70}$ | $78.01_{\pm 1.77}$ |
| BQN | $65.49_{\pm 1.92}$ | $51.76_{\pm 2.73}$ | $30.28_{\pm 3.32}$ | $35.49_{\pm 2.77}$ | $80.83_{\pm 1.16}$ | $67.23_{\pm 3.20}$ | $50.42_{\pm 6.64}$ | $26.48_{\pm 7.52}$ | $33.95_{\pm 6.80}$ | $76.13_{\pm 1.62}$ |
| BQN+HDM | $68.53_{\pm 0.88}$ | $54.57_{\pm 1.09}$ | $31.03_{\pm 0.47}$ | $36.22_{\pm 0.61}$ | $81.87_{\pm 0.37}$ | $70.60_{\pm 2.34}$ | $57.50_{\pm 3.75}$ | $29.81_{\pm 2.62}$ | $33.14_{\pm 2.45}$ | $78.92_{\pm 1.25}$ |
| MLP | $66.34_{\pm 1.66}$ | $51.59_{\pm 3.86}$ | $29.84_{\pm 2.05}$ | $34.22_{\pm 2.10}$ | $80.39_{\pm 1.46}$ | $67.41_{\pm 7.36}$ | $51.67_{\pm 9.45}$ | $28.04_{\pm 6.46}$ | $28.84_{\pm 4.93}$ | $76.31_{\pm 2.55}$ |
| BRep(MLP+HDM) | $69.31_{\pm 1.32}$ | $55.80_{\pm 2.16}$ | $32.45_{\pm 1.30}$ | $37.52_{\pm 1.44}$ | $83.37_{\pm 0.87}$ | $72.22_{\pm 4.51}$ | $59.58_{\pm 5.37}$ | $28.97_{\pm 1.98}$ | $31.36_{\pm 1.95}$ | $79.72_{\pm 1.04}$ |

## C.2. Processing of Differential Time Series

To compute the differential curves for the three models (Original, FSTA-EC, and BRep), the original time series are split into ASD and NC groups: $T_{\mathrm{ASD}} \in \mathbb{R}^{b_1 \times n \times t}$ and $T_{\mathrm{NC}} \in \mathbb{R}^{b_2 \times n \times t}$, where $b_1$, $b_2$ denote the numbers of ASD and NC subjects, $n$ denotes the number of ROIs, and $t$ stands for the dimension of each time series. For each model, the processed time series are $T_*^{\mathrm{ASD}} \in \mathbb{R}^{b_1 \times n \times t}, T_*^{\mathrm{NC}} \in \mathbb{R}^{b_2 \times n \times t}$, where $*$ can be *org, FSTA-EC, BRep*, standing for the original time series, FSTA-EC processed and BRep processed time series, respectively. Group-level templates are obtained by averaging across subjects: $T_{\mathrm{ASD}}^{\mathrm{Template}\,*} = \frac{1}{b_1} \sum_{i=1}^{b_1} T_*^{\mathrm{ASD}}{}_i \in \mathbb{R}^{n \times t}, T_{\mathrm{NC}}^{\mathrm{Template}\,*} = \frac{1}{b_2} \sum_{i=1}^{b_2} T_*^{\mathrm{NC}}{}_i \in \mathbb{R}^{n \times t}$. Finally, group-level templates are averaged over ROIs, and their difference yields the differential time series template for each model, that is, $T_{\mathrm{differential}}^{\mathrm{Template}\,*} = \frac{1}{n} \sum_{i=1}^{n} \left( T_{\mathrm{ASD}}^{\mathrm{Template}\,*} \ominus T_{\mathrm{NC}}^{\mathrm{Template}\,*} \right)_i \in \mathbb{R}^t$.

## C.3. Visualization of Time Series Patterns between ASD and NC

Figs. 10 and 11 present the heatmaps of NC and ASD time series under different processing methods. Compared to the Original and FSTA-EC time series, the BRep-processed time series exhibit clearer and more structured patterns, with reduced noise and enhanced regularity. This suggests that BRep is able to effectively extract informative signals while suppressing irrelevant variations, demonstrating superior denoising capability and stronger representation learning capacity.

## C.4. Experiments on Multi-Class Brain Disorder Classification

Given that the main body of the paper evaluates only a limited set of datasets, additional experiments on the ADNI and PPMI multi-class datasets are included in this section.

Comparison of four baseline models and +HDM variants on ADNI, PPMI are shown in Tab. 3. The results in the table show that the +HDM variant outperforms the original model across nearly all metrics and architectures. Moreover, BRep achieves consistently strong performance on both datasets.

## C.5. Comparison Between BRep's Learned FC and Non-End-to-End FC Patterns

To compare BRep's learned FC with other non-end-to-end FC patterns, this section employs other methods' learned connectivity as input to the same MLP predictor (*i.e.*, GAT+MLP, Graphormer+MLP, and BRep+MLP), ensuring a fair and consistent evaluation. As shown in Fig. 12, the MLP equipped with BRep's learned FC achieves the best performance, indicating that the connectivity learned by BRep is more informative than that obtained by GAT or Graphormer. Note that GAT and Graphormer still rely on hand-crafted Pearson Correlation matrix with attention, whereas BRep learns FC directly from time series in an end-to-end fashion.

## C.6. Performance of BRep under linear and nonlinear settings

To test the performance benefits of BRep across both linear and nonlinear settings, we evaluate four combinations: (i) conventional Pearson correlation connectivities with a linear probe, (ii) Pearson with a nonlinear MLP, (iii) BRep connectivities with a linear probe, and (iv) BRep (with a nonlinear MLP). As Fig. 13 shows, in the linear probing setting, BRep consistently provides an advantage over conventional correlation connectivities on both ABIDE and ADHD-200, while its performance does not match that of the nonlinear MLPs. This indicates that the gain of BRep comes from the

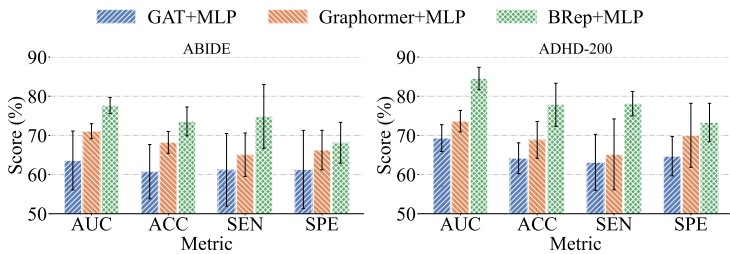

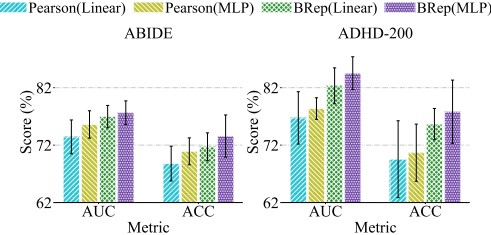

*Figure 12.* Comparison between BRep's FC and non-end-to-end FC patterns on ABIDE and ADHD-200.

*Figure 13.* Comparison of Pearson and BRep with linear and MLP predictors

*Table 4.* Performance comparison with graph structure learning (GSL) baselines on ABIDE and ADHD-200 ($mean_{\pm std}$). **Bold** and underlined indicate the best and second-best results, respectively.

| Type | Model | ABIDE | | | | ADHD-200 | | | |
|------|-------|-------|--|--|--|----------|--|--|--|
| | | AUC ↑ | ACC ↑ | SEN ↑ | SPE ↑ | AUC ↑ | ACC ↑ | SEN ↑ | SPE ↑ |
| *GSL* | Pro-GNN | $75.98_{\pm 1.23}$ | $71.36_{\pm 5.58}$ | $71.76_{\pm 9.59}$ | $67.87_{\pm 6.41}$ | $79.44_{\pm 1.76}$ | $\underline{74.31_{\pm 3.57}}$ | $74.43_{\pm 6.59}$ | $70.94_{\pm 6.94}$ |
| | IDGL | $\underline{73.19_{\pm 3.24}}$ | $\underline{70.01_{\pm 4.39}}$ | $68.11_{\pm 6.84}$ | $\underline{67.09_{\pm 2.37}}$ | $\underline{80.73_{\pm 2.74}}$ | $73.28_{\pm 4.53}$ | $72.37_{\pm 5.70}$ | $\underline{71.71_{\pm 4.35}}$ |
| | SLAPS | $72.82_{\pm 2.01}$ | $68.82_{\pm 3.68}$ | $\underline{71.97_{\pm 5.63}}$ | $65.14_{\pm 5.06}$ | $75.48_{\pm 4.62}$ | $71.62_{\pm 5.36}$ | $\underline{76.66_{\pm 10.30}}$ | $64.60_{\pm 9.04}$ |
| *Ours* | BRep | $\mathbf{77.64_{\pm 2.06}}$ | $\mathbf{73.58_{\pm 3.67}}$ | $\mathbf{74.86_{\pm 8.16}}$ | $\mathbf{68.12_{\pm 5.21}}$ | $\mathbf{84.53_{\pm 2.85}}$ | $\mathbf{77.82_{\pm 3.50}}$ | $\mathbf{78.11_{\pm 3.12}}$ | $\mathbf{73.30_{\pm 4.90}}$ |

learned BRep-based connectivity itself rather than merely from using a more expressive predictor.

## C.7. Comparison Between BRep and generic GSL methods

Tab. 4 compares BRep against generic GSL methods (Pro-GNN (Jin et al., 2020), IDGL (Chen et al., 2020), SLAPS (Fatemi et al., 2021)) under identical protocols. BRep consistently achieves the best performance. For example, on ADHD-200, BRep improves ACC by $3.51\%$ over the best GSL baseline (Pro-GNN). Theoretically, this is because a simple inner-product lacks the expressivity to capture continuous high-order dependencies (Theorem 2.4), whereas HDM guarantees universal approximation (Theorems 2.1 & 2.6). These results empirically and theoretically prove that the gain stems directly from HDM's superior capability in capturing high-order temporal dependencies.

## C.8. Hyperparameter Analysis.

This experiment aims to examine the impact of certain parameters on model performance. Four hyperparameter sensitivity experiments are as follows.

*(1) Number of HDM layers.* Fig. 14 shows that a single-layer HDM achieves the best performance, while deeper HDMs (2–5 layers) result in slightly lower but stable outcomes. This suggests that excessive depth does not benefit representation learning in this setting.

*(2) TopK connection.* As illustrated in Fig. 15, performance generally improves as $k$ increases from 20 to 80, with the best results observed at $k = 80$ on both datasets. This highlights the importance of appropriate graph sparsification for constructing reliable functional connectivity matrices.

*(3) Denoising loss weight.* Fig. 16 shows that the optimal performance occurs when the denoising loss weight is set to 2. This validates the contribution of the denoising component and emphasizes the need for balancing reconstruction and classification objectives: too small a weight diminishes the denoising effect, whereas too large a weight overemphasizes it.

*(4) Noise ratio.* According to Fig. 17, model performance remains robust when the noise ratio varies between 0.05 and 0.2. On ADHD-200, the best performance is observed at 0.05, while on ABIDE, results are largely stable across all noise levels. This demonstrates the robustness of the model in reconstructing time series even under moderate noise perturbations.

## D. Dimensional constraints analysis with varying time series lengths

For datasets with heterogeneous durations, standard sequence processing techniques can be seamlessly applied: longer sequences can be adapted via sampling or truncation, while shorter sequences can be padded (*e.g.*, zero-padding) or

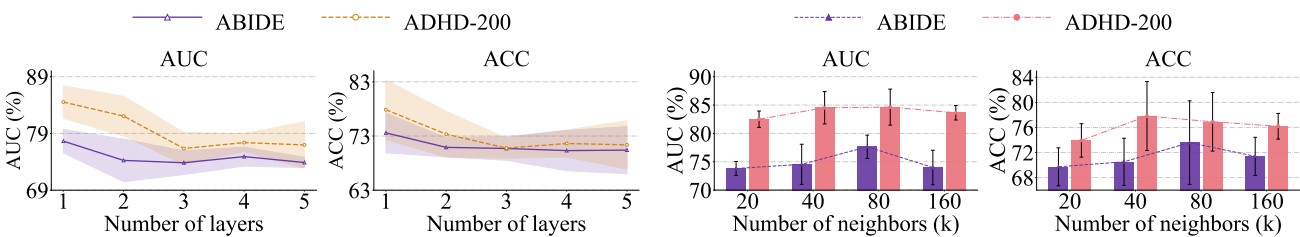

Figure 14. Impact of the number of HDM layers.

Figure 15. Impact of $K$ in TopK.

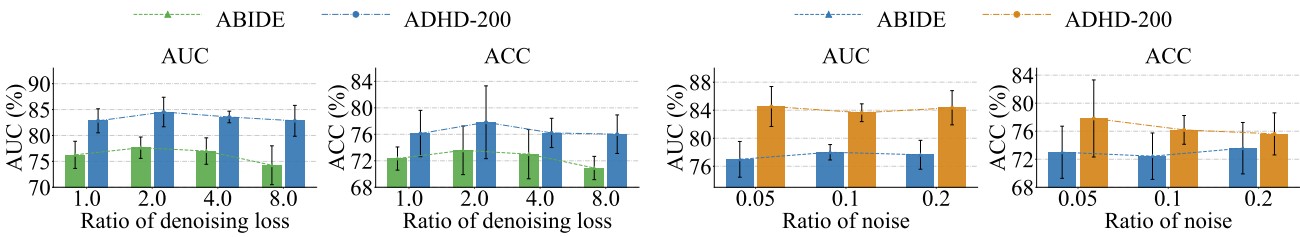

Figure 16. Impact of the denoising loss weight.

Figure 17. Impact of the noise ratio.

Table 5. Performance comparison on ABIDE and ADHD-200 with randomly sampled time series of length $D = 80$ ($mean_{\pm std}$). **Bold** and underlined indicate the best and second-best results, respectively.

| Type | Model | ABIDE | | | | ADHD-200 | | | |
|------|-------|-------|-----|-----|-----|----------|-----|-----|-----|
| | | AUC ↑ | ACC ↑ | SEN ↑ | SPE ↑ | AUC ↑ | ACC ↑ | SEN ↑ | SPE ↑ |
| *GNNs* | A-GCL | $73.86_{\pm 2.91}$ | $71.04_{\pm 2.40}$ | $71.42_{\pm 3.03}$ | $70.95_{\pm 3.19}$ | $74.78_{\pm 4.39}$ | $73.11_{\pm 4.30}$ | $72.04_{\pm 4.68}$ | $\underline{73.08_{\pm 4.10}}$ |
| *GTs* | BrainNETTF | $77.93_{\pm 1.41}$ | $69.26_{\pm 2.26}$ | $65.92_{\pm 8.60}$ | $\mathbf{73.20_{\pm 6.06}}$ | $79.79_{\pm 3.14}$ | $72.67_{\pm 3.17}$ | $73.64_{\pm 11.06}$ | $72.08_{\pm 5.66}$ |
| | ALTER | $\underline{77.99_{\pm 2.21}}$ | $70.10_{\pm 2.26}$ | $72.84_{\pm 7.40}$ | $67.68_{\pm 5.81}$ | $\underline{83.16_{\pm 1.61}}$ | $73.48_{\pm 1.38}$ | $74.58_{\pm 6.85}$ | $72.20_{\pm 5.82}$ |
| | BioBGT | $69.96_{\pm 1.18}$ | $69.70_{\pm 2.92}$ | $67.04_{\pm 3.41}$ | $72.02_{\pm 4.67}$ | $71.64_{\pm 1.14}$ | $71.06_{\pm 0.08}$ | $75.39_{\pm 5.45}$ | $71.92_{\pm 2.29}$ |
| *NNs* | MLP | $75.60_{\pm 2.38}$ | $70.92_{\pm 2.34}$ | $63.96_{\pm 9.58}$ | $73.03_{\pm 7.68}$ | $78.36_{\pm 1.88}$ | $70.68_{\pm 4.98}$ | $74.15_{\pm 4.63}$ | $64.42_{\pm 7.60}$ |
| | BQN | $\mathbf{79.85_{\pm 1.27}}$ | $72.53_{\pm 1.41}$ | $73.26_{\pm 5.99}$ | $\underline{72.03_{\pm 6.24}}$ | $83.34_{\pm 1.13}$ | $75.68_{\pm 1.95}$ | $\mathbf{79.73_{\pm 2.27}}$ | $71.63_{\pm 4.87}$ |
| *Ours* | BRep(D=80) | $77.27_{\pm 0.93}$ | $71.12_{\pm 2.19}$ | $70.79_{\pm 8.97}$ | $68.88_{\pm 6.98}$ | $\underline{84.28_{\pm 2.35}}$ | $\underline{77.31_{\pm 1.75}}$ | $78.02_{\pm 5.89}$ | $72.74_{\pm 6.48}$ |
| | BRep | $77.64_{\pm 2.06}$ | $\mathbf{73.58_{\pm 3.67}}$ | $\mathbf{74.86_{\pm 8.16}}$ | $68.12_{\pm 5.21}$ | $\mathbf{84.53_{\pm 2.85}}$ | $\mathbf{77.82_{\pm 3.50}}$ | $\underline{78.11_{\pm 3.12}}$ | $\mathbf{73.30_{\pm 4.90}}$ |

interpolated to match the predefined dimension $D$. As shown in Tab. 5, BRep($D = 80$) demonstrates exceptional robustness. Despite halving the ADHD-200 sequence length (from 160 to 80), the performance drop is marginal. Remarkably, BRep($D = 80$) still outperforms or highly competes with all baseline models, even though those baselines utilized the full-length time series for Pearson correlation computation.

# E. Complexity analysis of BRep

Theoretically, the construction of traditional Pearson involves computing the inner product of normalized time series, requiring $O(N^2D)$ time and $O(N^2 + ND)$ memory. And the construction of HDM involves a linear projection $Z = XO$ and a subsequent graph construction $A = ZZ^\top$. The time complexity is $O(N^2D + ND^2)$, and the memory complexity is $O(N^2 + ND + D^2)$. The additional overhead compared to Pearson is bounded by $O(ND^2)$ in time and $O(D^2)$ in memory. For brain analysis tasks taking BOLD time series as input, both $N$ and $D$ are typically bounded and of similar moderate magnitudes (*e.g.*, $N = 200$, $D = 100$ in ABIDE; $N = 190$, $D = 160$ in ADHD-200). Thus, the $O(ND^2)$ and $O(D^2)$ terms do NOT constitute a computational bottleneck.

# F. Stability analysis of BRep

To verify the stability of BRep, this section randomly initialized 8 models and calculated the group-mean connection matrices of NC, ASD and their difference, as shown in Fig. 18. Moreover, the Pearson correlation and cosine similarity

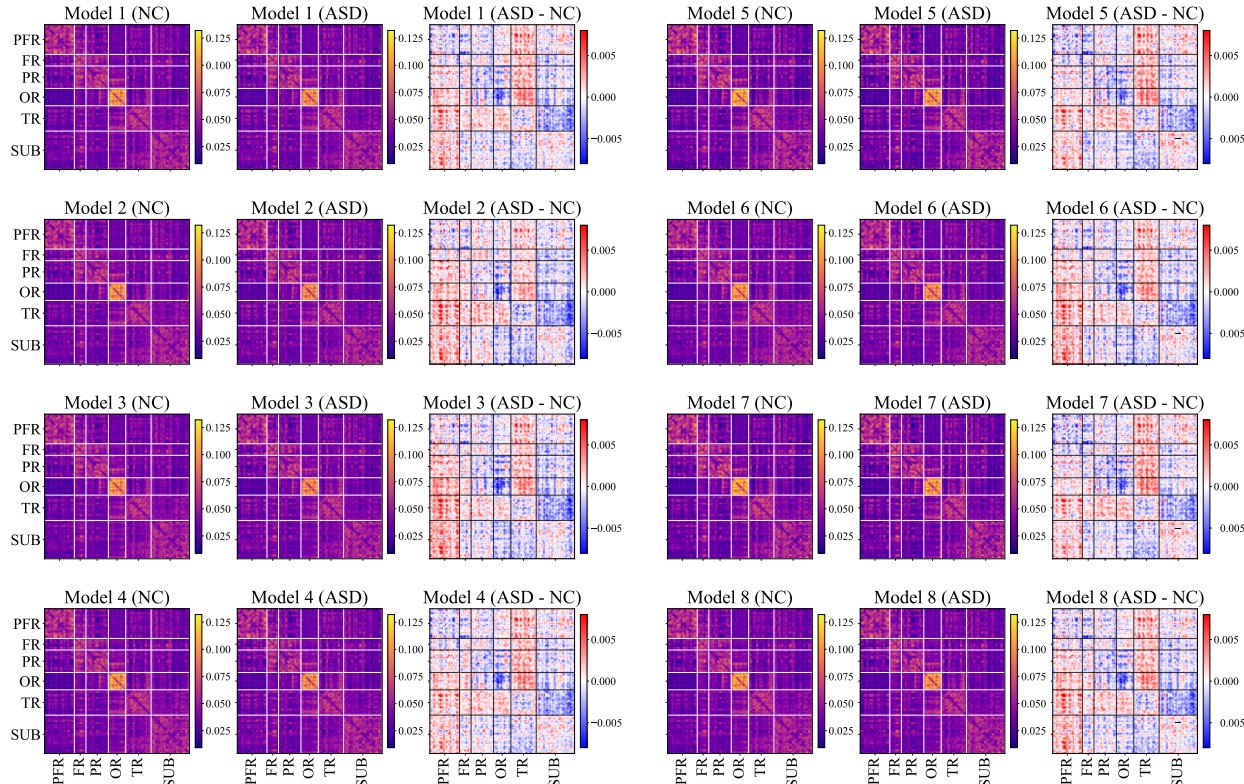

*Figure 18.* BRep-learned NC, ASD, difference connectivity matrices from 8 randomly initialized models.

*Table 6.* Group-mean graph stability measured by Pearson correlation and cosine similarity on ABIDE ($mean_{\pm std}$).

| Group | Pearson correlation (%) | Cosine similarity (%) |
|---|---|---|
| NC | $99.74_{\pm 0.15}$ | $99.82_{\pm 0.15}$ |
| ASD | $99.77_{\pm 0.14}$ | $99.82_{\pm 0.17}$ |
| ASD-NC | $90.30_{\pm 4.53}$ | $90.29_{\pm 4.53}$ |

of the connectivity matrices of these 8 models were calculated, as shown in Tab. 6. The $> 99.7\%$ similarity for NC/ASD confirms that BRep stably captures patterns regardless of initialization. The $> 90\%$ similarity in the differential networks (ASD-NC) proves consistent biomarker extraction.

## G. Performance variance analysis and optimal condition analysis

To conduct a detailed analysis of the conditions for the optimal and suboptimal results, Fig. 19 shows the boxplots and SEN-vs-SPE scatter plots of 20 independent runs with varying random seeds on both ABIDE and ADHD-200.

**Empirical Variance Analysis.** Fig. 19 (a) and b reveal that overall predictive capacity (AUC and ACC) remains highly stable, confirming that BRep consistently learns robust representations. However, the class-specific metrics (SEN and SPE) exhibit higher variance with a clear negative correlation (trade-off), as shown in Fig. 19 (c) and (d).

**Mechanistic Cause.** On small medical datasets, varying random initializations slightly alter the optimization trajectory of the global matrix O. Consequently, the strict TopK sparsification causes minor shifts in the model's implicit decision boundary. The selected subset graphs may occasionally become slightly biased toward the connectivity patterns of one specific class, driving the observed SEN/SPE trade-off.

Thus, the conditions for the model's performance states can be defined as follows:

- **Optimality (Balanced Topology)**. The model achieves optimal performance (*i.e.*, points clustered near the top-right

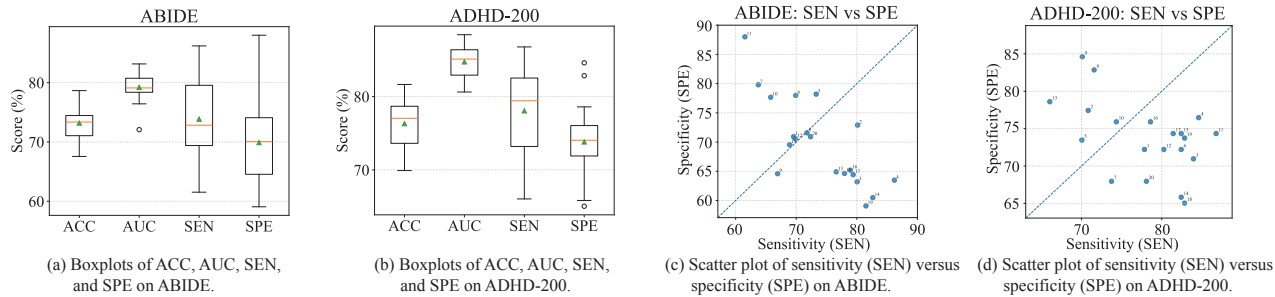

(a) Boxplots of ACC, AUC, SEN, and SPE on ABIDE.

(b) Boxplots of ACC, AUC, SEN, and SPE on ADHD-200.

(c) Scatter plot of sensitivity (SEN) versus specificity (SPE) on ABIDE.

(d) Scatter plot of sensitivity (SEN) versus specificity (SPE) on ADHD-200.

*Figure 19.* Boxplots and SEN-vs-SPE scatter plots of 20 independent runs with varying random seeds on both ABIDE and ADHD-200.

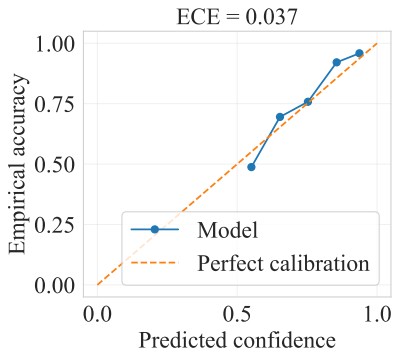

*Figure 20.* Reliability Diagram and ECE on ABIDE

diagonal of the scatter plots) when the optimization trajectory successfully captures shared topological biomarkers for both classes. This balance is typically secured when the denoising regularization term appropriately acts as a constraint, preventing the graph structure from overfitting to class-specific noise.

- **Sub-optimality (Topological Bias)**. Sub-optimal runs (*i.e.*, points at the extreme top-left or bottom-right of the scatter plots) occur when the learnable parameter $O$ falls into a local minimum that heavily favors one class's connectivity patterns. While this maximizes the accuracy for one specific group (yielding high SEN or high SPE), it degrades the model's ability to generalize to the other group, resulting in unbalanced overall performance.

## H. Risk Calibration Analysis

To evaluate whether the predicted probabilities are reliable at the individual level, this study conducts a risk calibration analysis. A reliability diagram is generated using the model's predicted confidence scores, and the Expected Calibration Error (ECE) is computed to quantify calibration performance. As shown in Fig. 20, the calibration curve closely aligns with the ideal diagonal line across the full confidence range, indicating strong consistency between the predicted probabilities and the empirical positive rates. On the ABIDE, the model achieves ECE of 0.037, which demonstrates excellent calibration quality and indicates that the predicted risk scores accurately reflect each subject's true likelihood of disease.

## I. Proofs of Theorems

### I.1. Proof of Theorem 2.1

*Proof.* The map $S : (\mathbf{x}, \mathbf{y}) \mapsto (\mathbf{u}, \mathbf{v}) = (\mathbf{W}\mathbf{x}, \mathbf{W}\mathbf{y})$ is a linear homeomorphism of $\mathbb{R}^{2m}$ because $\mathbf{W}$ is invertible; in particular $S$ restricts to a homeomorphism $S : K \to S(K)$ and $S(K)$ is compact. The mapping

$$(\mathbf{x}, \mathbf{y}) \mapsto (\mathbf{u}, \mathbf{v}, \text{vec}(\mathbf{u}\mathbf{v}^\top))$$

is continuous (it is a composition of linear maps and polynomial multiplications); denote it by $T$. Since $S$ is injective on $K$ and $T$ includes $(\mathbf{u}, \mathbf{v})$ as components, $T$ is injective on $K$. Hence $T : K \to T(K)$ is a homeomorphism onto its compact image $T(K)$, and the inverse $T^{-1} : T(K) \to K$ is continuous.

Define $g_0 := h \circ T^{-1} : T(K) \to \mathbb{R}$. Then $g_0$ is continuous on the compact set $T(K)$. By the universal approximation theorem, there exists a network $g \in \mathcal{H}$ such that

$$\sup_{\mathbf{z} \in T(K)} |g_0(\mathbf{z}) - g(\mathbf{z})| < \varepsilon,$$

which implies for all $(\mathbf{x}, \mathbf{y}) \in K$,

$$|h(\mathbf{x}, \mathbf{y}) - g(T(\mathbf{x}, \mathbf{y}))| = |g_0(T(\mathbf{x}, \mathbf{y})) - g(T(\mathbf{x}, \mathbf{y}))| \leq \sup_{\mathbf{z} \in T(K)} |g_0(\mathbf{z}) - g(\mathbf{z})| < \varepsilon.$$

This proves the claim. $\qquad\square$

### I.2. Proof of Theorem 2.4

*Proof.* Fix some $\mathbf{y}_0$ such that $\{(\mathbf{x}, \mathbf{y}_0) : \mathbf{x} \in U\} \subset K$ with $U$ containing two points $\mathbf{x}^{(1)} \neq \mathbf{x}^{(2)}$ as in the statement. Suppose by contradiction that $\mathcal{F}_1$ were dense in $C(K)$. Consider the continuous function $h(\mathbf{x}, \mathbf{y}) := \mathbf{x}_1^2$ restricted to $K$. For any $M \in \mathbb{R}^{m \times m}$ and $\phi \in C(\mathbb{R})$ define $f(\mathbf{x}, \mathbf{y}) := \phi(\mathbf{x}^\top M \mathbf{y})$. Fixing $\mathbf{y} = \mathbf{y}_0$, the function $\mathbf{x} \mapsto f(\mathbf{x}, \mathbf{y}_0)$ depends on $\mathbf{x}$ only through the scalar linear projection $\ell(x) := \mathbf{x}^\top (M\mathbf{y}_0)$. Thus for any two distinct $\mathbf{x}^{(1)}, \mathbf{x}^{(2)}$ in $U$ satisfying $\ell(\mathbf{x}^{(1)}) = \ell(\mathbf{x}^{(2)})$ we have $f(\mathbf{x}^{(1)}, \mathbf{y}_0) = f(\mathbf{x}^{(2)}, \mathbf{y}_0)$ while $h(\mathbf{x}^{(1)}, \mathbf{y}_0) \neq h(\mathbf{x}^{(2)}, \mathbf{y}_0)$ provided we choose $\mathbf{x}^{(1)}, \mathbf{x}^{(2)}$ so that $\mathbf{x}_1^{(1)} \neq \mathbf{x}_1^{(2)}$.

Because the kernel of a nontrivial linear functional has positive codimension, for any fixed $M$ and $\mathbf{y}_0$ one can find distinct $\mathbf{x}^{(1)}, \mathbf{x}^{(2)}$ with $\ell(\mathbf{x}^{(1)}) = \ell(\mathbf{x}^{(2)})$ and yet $\mathbf{x}_1^{(1)} \neq \mathbf{x}_1^{(2)}$ (for example, take $\mathbf{x}^{(2)} = \mathbf{x}^{(1)} + \mathbf{z}$ with $\mathbf{z} \in \ker(M\mathbf{y}_0)$ but $\mathbf{z}_1 \neq 0$; such $\mathbf{z}$ exists for generic choices because $\ker(M\mathbf{y}_0)$ is either of dimension $\geq 1$ or the restriction on $U$ ensures existence). Thus for that $M$ and any $\phi$ we have

$$\sup_{\mathbf{x} \in U} |h(\mathbf{x}, \mathbf{y}_0) - f(\mathbf{x}, \mathbf{y}_0)| \geq |h(\mathbf{x}^{(1)}, \mathbf{y}_0) - f(\mathbf{x}^{(1)}, \mathbf{y}_0)| + |h(\mathbf{x}^{(2)}, \mathbf{y}_0) - f(\mathbf{x}^{(2)}, \mathbf{y}_0)|$$

giving a positive lower bound. Since this holds for each $M$ the family $\mathcal{F}_1$ cannot approximate $h$ uniformly within arbitrarily small error; hence $\mathcal{F}_1$ is not dense in $C(K)$. $\qquad\square$

### I.3. Proof of Theorem 2.6

*Proof.* Direct computation yields

$$s_{ij}(\mathbf{x}, \mathbf{y}) = \mathbf{x}_i \mathbf{y}_j,$$

so $s(\mathbf{x}, \mathbf{y}) = \text{vec}(\mathbf{x}\mathbf{y}^\top)$ by definition of vectorization. The mapping $(\mathbf{x}, \mathbf{y}) \mapsto \text{vec}(\mathbf{x}\mathbf{y}^\top)$ is polynomial and hence continuous. On any domain where distinct $(\mathbf{x}, \mathbf{y})$ lead to distinct outer-products $\mathbf{x}\mathbf{y}^\top$ (for instance if $K$ does not identify distinct pairs through the same outer-product), the mapping is injective. Because $K$ is compact, the image $s(K)$ is compact and $s$ is a homeomorphism onto its image when restricted to a region of injectivity. Define $g_0 := h \circ s^{-1}$ on $s(K)$; $g_0$ is continuous on the compact set $s(K)$. The universal approximation theorem produces an MLP $g$ approximating $g_0$ uniformly on $s(K)$ to within $\varepsilon$. Pulling back yields the desired uniform approximation of $h$ on $K$ by $g \circ s$. $\qquad\square$

## J. Limitations and Future Work

Despite its superior performance, BRep has a few limitations to address in future work:

- **Pairwise Connectivity:** BRep captures high-order temporal dependencies between ROI pairs. Since cognitive processes often involve synergistic interactions among multiple regions, extending to hypergraph or tensor-based representations could further capture these complex dynamics.

- **Fixed Input Dimension:** The learnable matrix $O$ requires a fixed sequence length $D$. While standard truncation and padding work empirically well (Appendix D), this constraint requires preprocessing alignment for multi-site datasets with severe duration heterogeneity.

