# OpenReview forum: "Brain Networks Should Be Learned, Not Constructed"
_ICML.cc/2026/Conference — ICML 2026 regular_

### Official Review · Reviewer_umHi · 2026-03-03

**Soundness:** 2
**Presentation:** 3
**Significance:** 3
**Originality:** 2
**Overall Recommendation:** 4
**Confidence:** 3

**Summary:**

This paper attempt to address the limitation of separating brain network construction from downstream analysis by proposing an end-to-end framework called BRep. The authors introduce a parametric High-order Dependence Measure to learn learnable graph structures directly from BOLD time series, replacing fixed hand-crafted correlations like Pearson correlation. This measure is theoretically grounded in U/V-statistics and is implemented using a learnable bilinear transformation followed by TopK sparsification. To prevent the learned graph from collapsing into trivial solutions, the framework incorporates a denoising regularization task where a Graph Neural Network reconstructs corrupted signals using the learned connectivity. By optimizing the graph structure itself, the authors show that a simple MLP predictor can achieve superior performance compared to complex Graph Neural Networks and Graph Transformers on benchmarks such as ABIDE and ADHD-200.

**Compliance With Llm Reviewing Policy:**

Affirmed.

**Final Justification:**

The authors' rebuttal provided empirical evidence addressed my primary concerns, these practical strengths outweigh my remaining methodological reservations, justifying an increase in my score to a 4 (Weak Accept).

**Key Questions For Authors:**

Referring to the weaknesses.

**Limitations:**

No.

The authors have not explicitly discussed the limitations of their proposed framework in the main text. I suggest the authors include a discussion on the following points:
1. Rigidity of Input Dimension: The learnable matrix $O$ requires a fixed time-series length ($D$), which limits the model's flexibility when handling multi-site fMRI datasets with varying scan durations.
2. Hyperparameter Sensitivity: As indicated in Appendix C.7 (Fig. 16), the model's performance fluctuates significantly with the denoising loss weight $\lambda$. This sensitivity poses a challenge for practical deployment where optimal tuning might be difficult.
3. Temporal Assumption: The method implicitly assumes temporal alignment across subjects (due to the shared transformation $O$), which is generally not the case for resting-state fMRI data.

**Strengths And Weaknesses:**

Strengths:
1. Relevance to Domain: The paper attempt to address a significant bottleneck in brain network analysis: the reliance on fixed, hand-crafted correlation measures (e.g., Pearson correlation). Proposing an end-to-end graph learning framework to replace the preprocessing step is a meaningful direction for the field.
2. Efficiency: The proposed BRep framework demonstrates that by learning a better graph structure, the downstream predictor can be simplified (e.g., a simple MLP) while still achieving competitive results. This offers a potential solution to the over-smoothing and computational complexity issues associated with deep GNNs.
3. Theoretical Grounding: The authors attempt to ground their "High-order Dependence Measure" (HDM) in U/V-statistics theory. While the implementation simplifies to a bilinear form, providing a theoretical justification for the learnable correlation adds value to the presentation.
4. Interpretability: The visualization of the learned adjacency matrices (e.g., differences between ASD and NC groups) shows better modularity and cleaner structures compared to generative baselines like VGAE, aligning well with neuroscientific expectations.

Weaknesses:
1. Differentiability of TopK: A critical technical detail is missing. The authors use a TopK operation in Eq. (7) to sparsify the graph within an end-to-end training pipeline. Standard TopK operations are non-differentiable with respect to the sorting indices. The paper does not explain how gradients are propagated through this step. Without this, the mechanism for updating the parameter matrix $O$ via the classification loss is mathematically unclear and raises reproducibility concerns.
2. Rigidity of Matrix $O$: The learnable weight matrix $O$ is defined with dimensions $D \times D$, where $D$ is the length of the time series. This design is highly rigid. Resting-state fMRI scans often vary in duration across datasets and sites. Furthermore, rs-fMRI data is generally not time-locked across subjects; the spontaneous fluctuations are random. Learning a fixed transformation on the temporal axis implies a temporal consistency across subjects that does not biologically exist.
3. Fairness of Baselines: The paper positions itself as a "Graph-structured Brain Representation Learning" method, effectively a Graph Structure Learning (GSL) approach. However, the baselines provided (e.g., BrainGNN, BrainGB, Graphormer) all rely on fixed graph structures constructed via hand-crafted correlations (Pearson). The authors fail to compare BRep against existing generic GSL methods (e.g., IDGL, ProGNN, SLAPS) or even a simple end-to-end learnable similarity baseline (e.g., learnable cosine similarity + GNN). Comparing a learnable graph method only against fixed graph methods  proves that learning a graph is better than fixing a graph, but it does not prove that the proposed High-order Dependence Measure is superior to other graph learning techniques.
4. Originality & Presentation: While the application to fMRI is novel, the core technical contribution—a learnable bilinear similarity ($Z Z^T$) followed by sparsification—is technically a variation of self-attention mechanisms widely used in graph structure learning (GSL). The framing of this mechanism as a "High-order Dependence Measure" based on U-statistics, while theoretically interesting, feels like an over-packaging of a standard bilinear attention mechanism.

---

> ### Author Rebuttal · Authors · 2026-03-31
>
> > W1. Differentiability of TopK
>
> Please refer to the response R1 for Reviewer xLXn.
>
>
>
> > W2. Rigidity of matrix and biological validity of temporal transformation
>
> R2. We sincerely thank the reviewer for the insightful observation. We completely agree with the biological premise that resting-state fMRI spontaneous fluctuations are not time-locked across subjects.
>
> However, the design of our learnable matrix actually conforms to standard data processing practices widely adopted in this field. Typically, to handle datasets where resting-state fMRI scans vary in duration across individuals, the standard practice is to truncate the sequences of different subjects to match the shortest one.
>
> More importantly, our model is fundamentally designed to capture the correlations between different brain regions *within the same individual*. Therefore, as long as the signal acquisitions of different brain regions within a single subject are temporally aligned, the model functions exactly as intended. It absolutely does not require, nor does it assume, any temporal alignment or consistency *across different individuals*.
>
> Based on this specific characteristic of our model, processing inputs of varying lengths within the same dataset is straightforward and highly flexible: longer sequences can be truncated via sampling, and shorter sequences can simply be padded. We will explicitly clarify our adherence to standard intra-subject alignment and this preprocessing flexibility in the revised manuscript to resolve this misunderstanding.
>
> **The empirical validation of this strategy on clinical datasets is detailed in our response R5 to Reviewer YaUL**
>
>
>
> > W3. Fairness of Baselines and New Comparison
>
> R3. We clarify the baseline selection and provide the requested comparisons below:
>
> While recent brain network models have explored graph structure learning, they predominantly remain anchored to hand-crafted correlations (e.g., Pearson) at various stages of their pipelines. For instance, although FBNetGen generates graphs from time series, it still relies on Pearson correlation matrices as node features. Similarly, attention-based graph learners like BioBGT rely on pre-computed node features rather than raw time series. Thus, our initial evaluation focused on demonstrating that BRep can completely eliminate the reliance on Pearson-derived features, replacing these established pipelines with a unified end-to-end representation.
>
> As requested, we have compared BRep against generic GSL methods (Pro-GNN, IDGL, SLAPS) under identical protocols. The results are summarized **[Results](https://anonymous.4open.science/r/BRep-demo-1A3E/fig/GSL.png)**. BRep consistently achieves the best performance. For example, on ADHD-200, BRep improves ACC by 3.51% over the best GSL baseline (Pro-GNN). Theoretically, this is because a simple inner-product lacks the expressivity to capture continuous high-order dependencies (Theorem 2.4), whereas HDM guarantees universal approximation (Theorems 2.1 & 2.6). These results empirically and theoretically prove that the gain stems directly from HDM's superior capability in capturing high-order temporal dependencies.
>
>
>
> > W4. The framing of this mechanism as a "High-order Dependence Measure" based on U-statistics, while theoretically interesting, feels like an over-packaging of a standard bilinear attention mechanism.
>
> R4. There must be some serious misunderstandings. The core contribution is **NOT** the form of the learnable bilinear similarity $ZZ^{\top}$ but **the justification of its capability on measuring "High-order Dependence” between two time series.**, which is not trivial and obvious. The main challenge stems from the fact that $x_i = (x_{i1}, x_{i2}, \cdots, x_{iD})$ is the **time series** captured from the i-th ROI (Line 082) instead of the feature of i-th ROI. Since each element $x_{ij}$ is just a signal snapshot without any specific meaning, it doesn’t make sense to directly feed it into the MLP as $z_i = x_i O$ (Line 183-186). The main technical contribution of this paper is to justify that the learnable bilinear similarity $z_i z_j^{\top}$ can measure "High-order Dependence” between two time series $x_i$ and $x_j$. To this end, the learnable bilinear similarity $z_i z_j^{\top}$ is firstly extended from Linear and Nonlinear Correlations (Section 2.2) and then theoretically guaranteed to approximate the higher-order statistics (Section 2.4). By regarding $z_i z_j^{\top}$ as a bilinear attention mechanism, our contribution can be seen as **justifying the ability of a single attention operation on measuring High-order Dependence**. We believe it is a significant and novel understanding of the attention mechanism instead of an over-packaging of a standard bilinear attention mechanism.
>
>
>
> > Limitation
>
> Thank you for this valuable comment. We will further discuss these limitations in the revised manuscript.

---

> > ### Author Rebuttal · Reviewer_umHi · 2026-04-02
> >
> > I thank the authors for their detailed rebuttal and the substantial effort in conducting the new experiments. The inclusion of generic GSL baselines (e.g., Pro-GNN, IDGL) effectively addresses my primary concern regarding the fairness of the evaluation. While I still have minor reservations about the theoretical framing and the hard-masking nature of the TopK gradients, the empirical results presented in the rebuttal are convincing. Despite these remaining limitations, I am pleased to see a practical new approach introduced to the field of fMRI data analysis. In light of these new findings, I will raise my score to 4. Please ensure that these additional baselines and implementation details are fully incorporated into the final version of the manuscript.

---

### Official Review · Reviewer_xLXn · 2026-03-08

**Soundness:** 3
**Presentation:** 3
**Significance:** 4
**Originality:** 4
**Overall Recommendation:** 5
**Confidence:** 5

**Summary:**

This paper presents an end-to-end Brain Representation framework BRep that discarding fixed, hand-crafted correlation matrices. Instead, it directly maps BOLD time series to a dynamic graph-structured representation using a learnable, high-order parametric estimator. The framework employs a simplified MLP predictor, constrained by a GCN-based denoising autoencoder for regularization. Backed by mathematical proofs claiming the universal approximation of high-order U/V-statistics, the authors report that this streamlined approach outperforms established baselines across multiple datasets.

**Compliance With Llm Reviewing Policy:**

Affirmed.

**Final Justification:**

All the concerns have been stressed. I will keep my positive score.

**Key Questions For Authors:**

(i)The paper uses a TopK selection operator for sparse adjacency matrix construction but fails to explicitly detail gradient propagation through this non-differentiable operator during end-to-end training. Clarify how end-to-end training is enabled; if relaxation or surrogate gradients are used, provide specific details. (ii) The paper does not provide theoretical justification for why GCN-based corrupted BOLD signal reconstruction specifically improves the performance of the downstream MLP, nor for this coupling mechanism. (iii) A rigorous analysis of performance variance across metrics is lacking. The authors should clarify the conditions under which the model achieves optimality or sub-optimality.

**Limitations:**

yes

**Strengths And Weaknesses:**

**Strengths:**
(i) Graph structure learning is reasonably applied to brain analysis and has research merit. (ii) Theoretical derivation is complete with formal proofs and clear presentation. (iii) The model yields biologically interpretable results consistent with existing literature. Overall, the BRep framework is technically sound and empirically validated for functional brain network modeling.

**Weaknesses:**

(i)Some technical details (e.g., end-to-end training) are insufficiently explained. (ii) Although the marginal performance improvements in Table 2 are acceptable for these benchmarks, the paper lacks a rigorous analysis of performance variance across different metrics.

---

> ### Author Rebuttal · Authors · 2026-03-31
>
> > W1&Q1. End-to-end Training and Differentiability of TopK
>
> R1. End-to-end training is achieved via standard autograd through `torch.topk`, which is natively differentiable with respect to the selected values.
>
> In each forward pass, the TopK operator identifies the neighbor set $\Omega_i$ based on the similarity matrix $S$. We retain the exact continuous similarity values $S_{ij}$ for these selected indices and set others to zero:
> $$
> \tilde S_{ij}=
> \begin{cases}
> S_{ij}, & j\in\Omega_i\\\\
> 0, & j\notin\Omega_i
> \end{cases}
> $$
> During backpropagation, the autograd engine routes gradients exclusively through these retained continuous values. Unselected entries receive zero gradients:
> $$
> \frac{\partial \mathcal L}{\partial S_{ij}}=
> \begin{cases}
> \frac{\partial \mathcal L}{\partial \tilde S_{ij}}, & j\in\Omega_i \\\\
> 0, & j\notin\Omega_i
> \end{cases}
> $$
> The dynamic evolution of the graph structure is enabled by the globally shared parameter matrix $O$. Although gradients flow only through the active TopK edges, they update the global matrix $O$ used in the mapping $Z=XO$. This update shifts the entire latent representation $Z$ globally. Thus, in the subsequent forward pass, the pairwise similarities in $S=ZZ^{\top}$ are re-evaluated across all potential edges. This shared parameterization ensures the model can efficiently explore and update the graph topology in an end-to-end manner without requiring complex continuous relaxations.
>
> > W2. Performance variance and optimality conditions
>
> R2. To address this concern, we conducted 20 independent runs with varying random seeds on both ABIDE and ADHD-200. We summarize the variance using boxplots and SEN-vs-SPE scatter plots (shown in **[Results](https://anonymous.4open.science/r/BRep-demo-1A3E/fig/mix.png)**).
>
> + Empirical Variance Analysis. The boxplots reveal that overall predictive capacity (AUC and ACC) remains highly stable, confirming that BRep consistently learns robust representations. However, the class-specific metrics (SEN and SPE) exhibit higher variance with a clear negative correlation (trade-off).
> + Mechanistic Cause. On small medical datasets, varying random initializations slightly alter the optimization trajectory of the global matrix O. Consequently, the strict TopK sparsification causes minor shifts in the model's implicit decision boundary. The selected subset graphs may occasionally become slightly biased toward the connectivity patterns of one specific class, driving the observed SEN/SPE trade-off.
>
> Thus, we can define the conditions for the model's performance states:
>
> + Optimality (Balanced Topology): The model achieves optimal performance (i.e., points clustered near the top-right diagonal of the scatter plots) when the optimization trajectory successfully captures shared topological biomarkers for both classes. This balance is typically secured when the denoising regularization term appropriately acts as a constraint, preventing the graph structure from overfitting to class-specific noise.
> + Sub-optimality (Topological Bias): Sub-optimal runs (i.e., points at the extreme top-left or bottom-right of the scatter plots) occur when the learnable parameter $O$ falls into a local minimum that heavily favors one class's connectivity patterns. While this maximizes the accuracy for one specific group (yielding high SEN or high SPE), it degrades the model's ability to generalize to the other group, resulting in unbalanced overall performance.
>
> > Q3. DAE & Coupling Justification
>
> R3. In our model, the GCN-based reconstruction is not designed as a standalone task, but as a structure-aware denoising regularizer. By reconstructing perturbed BOLD time series, the model is encouraged to learn noise-robust representations. GCN is specifically adopted as the reconstructor because it explicitly leverages graph neighborhood information during recovery. Therefore, the auxiliary reconstruction objective preserves not only node-wise signal information but also structural consistency across ROIs. This graph-aware denoising mechanism is particularly suitable for BOLD data, where inter-regional dependencies are essential.
>
> Regarding the coupling mechanism, this design explicitly decouples structure learning from downstream prediction. While the auxiliary GCN ensures that the learned graph A captures topologically consistent BOLD signals (acting as a clean structural prior), the downstream MLP leverages this refined graph for prediction. Because the MLP does not perform iterative message passing, it prevents the recursive aggregation of residual noise, keeping the predictor simple, robust, and highly effective.

---

> > ### Author Rebuttal · Reviewer_xLXn · 2026-04-02
> >
> > All the concerns have been stressed. I will keep my positive score.

---

### Official Review · Reviewer_YaUL · 2026-03-09

**Soundness:** 2
**Presentation:** 3
**Significance:** 2
**Originality:** 2
**Overall Recommendation:** 4
**Confidence:** 4

**Summary:**

This paper investigates the limitations of traditional brain connectivity construction methods, which rely on predefined statistical measures such as Pearson correlation or other hand-crafted dependence estimators. These metrics are task-agnostic and limited in their ability to capture high-order dependencies in fMRI signals. To address this issue, the authors propose an end-to-end framework that replaces fixed connectivity measures with a learnable high-order dependence module (HDM), enabling adaptive and task-aware graph construction optimized for downstream classification. The paper further provides theoretical analysis to demonstrate the expressive power of the proposed formulation in approximating general high-order dependence functions, and reports empirical improvements over traditional connectivity-based approaches on benchmark neuroimaging datasets.

**Compliance With Llm Reviewing Policy:**

Affirmed.

**Final Justification:**

I commend the authors for a successful rebuttal, and am happy to increase my score by one point.

**Key Questions For Authors:**

1) Can the authors clarify more explicitly how the proposed HDM differs from existing learnable graph construction methods in graph neural networks or neuroimaging literature? What is the key methodological distinction?
2) The paper argues that HDM captures high-order dependence. Can the authors provide controlled experiments to demonstrate this advantage more directly?
3) What is the computational overhead of the proposed HDM compared to traditional correlation-based graph construction? Can the authors provide runtime and memory complexity analysis?
4) How stable are the learned connectivity structures across different random initializations or data splits? Do the learned graphs exhibit consistent patterns?
5) The paper employs TopK sparsification to construct the adjacency matrix. Whilst experiments analysed the impact of the TopK value, the TopK operation itself is discontinuous and non-differentiable. The paper should enhance the description of how to efficiently and stably handle this discontinuous operation during backpropagation.

**Limitations:**

No. The paper does not explicitly discuss its methodological limitations or potential societal implications. The authors are encouraged to briefly clarify the practical limitations of the proposed approach, such as generalization across datasets, computational overhead, and the gap between theoretical assumptions and real-world neuroimaging data.

**Strengths And Weaknesses:**

**Strengths:**

1) This paper replaces conventional predefined statistical connectivity measures with a learnable high-order dependence module, enabling end-to-end and task-aware graph construction. This provides a more flexible alternative to traditional functional connectivity estimation pipelines that are typically decoupled from downstream objectives.
2) The paper provides theoretical analysis to justify the expressive power of the proposed bilinear formulation.
3) The paper is generally well structured and clearly written. The motivation, moving from fixed statistical connectivity measures to learnable and task-aware graph construction, is easy to follow.

**Weaknesses:**

1) The methodological novelty appears limited. Learning graph structures or task-aware connectivity in an end-to-end manner has been explored in prior graph learning and neuroimaging works. The core contribution mainly replaces fixed similarity measures with a parameterized bilinear mapping, which may be viewed as an incremental extension rather than a fundamentally new paradigm.
2) The necessity of modeling “high-order dependence” is not rigorously validated. While the paper argues that traditional correlation measures are insufficient, it does not provide controlled experiments or synthetic benchmarks to explicitly demonstrate that the proposed HDM captures non-linear or higher-order dependencies that conventional methods fail to model.
3) Although the paper emphasizes brain network construction, it lacks in-depth analysis of the learned connectivity structures. There is limited investigation into whether the learned graphs align with known functional brain networks or exhibit meaningful neurobiological patterns. As a result, the practical interpretability and neuroscientific significance of the learned structures remain unclear.
4) The theoretical results mainly focus on universal approximation properties under idealized assumptions. However, the practical implementation relies on finite-dimensional parameterization and TopK sparsification. The gap between theoretical expressivity and practical constraints is not thoroughly analyzed, potentially limiting the strength of the theoretical justification.
5) Experiments show that the mapping dimension of HDM must be strictly aligned with the time series length to achieve optimal performance. This strong dependence on dimensional consistency may limit the model's flexibility when processing clinical data with varying sampling rates or lengths.

---

> ### Author Rebuttal · Authors · 2026-03-31
>
> > W1&Q1. Clarify HDM’s distinctions from existing graph learning
>
> R1. The novelties, including **novel problem** and **justifying old paradigm on new data type**, are remarkable.
>
> - **“Graph-structured brain representation (GBR)” and “Graph structure learning (GSL)” are different problems**, although both produce adjacency matrices. As shown in Fig. 1, **representation should simplify downstream predictors** by capturing rich semantics (Lines 067R), rather than relying on flexible models to process raw data. The GBR learned from BRep is directly fed into the **simple MLP** (Line 216 R). On the contrary, existing GSLs needs complex models (GNNs or GTs), since the learned graph is raw intermediate data and lacks sufficient semantics.
>
> - **Justifying parameterized bilinear mapping on time series** is the core technical contribution. It is not trivial on time series data. The challenge stems from that each $x_{ij}$ in $x_i = (x_{i1}, x_{i2}, \cdots, x_{iD})$ is just a signal snapshot, preveneting being fed into the MLP (Line 183 R). To overcome this, the bilinear similarity $z_i z_j^T$ is extended from Linear/Nonlinear Corr. (Sec. 2.2) and then theoretically guaranteed (Sec 2.4).
>
> > W2&Q2. Controlled experiments
>
> R2.
> **Benchmark Design:** A synthetic dataset containing 100 random variables (ROIs). We generated $X \sim {N}(0, 1)$ as an i.i.d. Gaussian noise sequence for ROI#1. For ROI#2, we generated a sequence $Y$ that strictly depends on the 3-order temporal joint state of $X$, formulated as $Y_t = X_t \cdot X_{t-1} \cdot X_{t-2} + \epsilon_t$, where $\epsilon_t \sim {N}(0, 0.01)$ is a minimal noise. All other 98 ROIs are pure noise.
>
> **Results:** We evaluated Pearson, dCor, and HDM by measuring the weight between ROI#1 ($X$) and ROI#2 ($Y$):
>
> - **Pearson/dCor:** Both of them failed to detect the relationship. The weights assigned to the $(X, Y)$ pair were $\approx 0.0$.
>
> - **HDM:** The HDM successfully assigned a **high weight** (approaching 1.0) between ROI#1 and ROI#2. This explicitly shows HDM's capability to capture complex interactions.
>
> > W3. The interpretability of the learned structures
>
> R3: Refer to **Sec. 3.2 (Interpretability Analysis)** and **Appx. C.1**. Main results are summarized as follows:
>
> * **Macro-anatomical Level:** As shown in Fig. 4, the learned graphs exhibit a clear modular structure, perfectly aligning with canonical brain network (Power et al., 2011).
> * **Subsystem & ROI Level:** We identified hyperconnectivity in the ACC and bilateral insula—core components as well as in the pSTS and MTG. They resonate with established literature (Uddin et al., 2013a; Supekar et al., 2013).
>
> > W4. The gap between theory and practice
>
> R4.
>
> - **Finite-dimensional cases.** Any results on finite-dimension in UAT can be employed here. Barron's Theorem shows that for any integer $N \ge 1$, there exists a single-hidden-layer NN $f_N(x)$ with $N$ neurons such that its mean squared error is bounded by $\mathbb{E}[(f(x) - f_N(x))^2] \le \frac{(2C_f)^2}{N}$, where the first moment of Fourier transform i.e., $C_f$, is bounded.
> - **TopK sparsification will NOT affect the results.** All the results are about the approximation of bilinear mapping, while the TopK is at the out of the bilinear mapping according to Eq.7.
>
> > W5. Dimensional constraints with varying lengths
>
> R5. Our design of fixing dimension $D$ for $O$ aligns with standard fMRI preprocessing pipelines, where varying lengths are unified to the minimum one.
>
> Because our model capture the dependencies between ROIs *within the same individual*, for datasets with heterogeneous durations, standard processing can be applied: longer sequences can be adapted via sampling or truncation, while shorter ones can be padded or interpolated to match the dimension $D$.
>
> As shown in the results ([Results](https://anonymous.4open.science/r/BRep-demo-1A3E/fig/sample.png)), despite halving the ADHD-200 length from 160 to 80, the performance drop is marginal. BRep ($D=80$) still outperforms or highly competes with all baselines (those with full-length input).
>
> > Q3. Complexity analysis
>
> R6. The overhead is by $O(ND^2)$ in time and $O(D^2)$ in memory. **$N$ and $D$ are of similar moderate magnitudes ($N=200$, $D=100$ in ABIDE). Thus, they do NOT cause a computational bottleneck**. We benchmarked the overhead as follows: [Result](https://anonymous.4open.science/r/BRep-demo-1A3E/fig/compl.png).
>
> > Q4. Stability of result
>
> R7. We computed the group-mean connectivity matrices and compared them across 8 models: [Result](https://anonymous.4open.science/r/BRep-demo-1A3E/fig/stab.png).
>
> The >$99.7\%$ similarity for NC/ASD confirms that BRep stably captures patterns regardless of initialization. The >$90\%$ similarity in the differential networks (ASD-NC) proves consistent  biomarkers extractions. Visualizations also confirm this [Results](https://anonymous.4open.science/r/BRep-demo-1A3E/fig/hm.pdf).
>
> > Q5. Differentiability of TopK
>
> R8. Refer to R1 for Reviewer xLXn.

---

> > ### Author Rebuttal · Reviewer_YaUL · 2026-04-02
> >
> > I have no more questions.

---

### Official Review · Reviewer_MjJS · 2026-03-12

**Soundness:** 3
**Presentation:** 2
**Significance:** 2
**Originality:** 2
**Overall Recommendation:** 4
**Confidence:** 4

**Summary:**

This paper addresses the limitations of traditional hand-crafted brain functional networks by proposing a learnable graph-structured representation. Unlike previous methods that rely on simple correlations, the proposed framework captures high-order dependencies between ROIs through an end-to-end learning process. Furthermore, the authors provide a theoretical analysis demonstrating that the prposed method can approximate arbitary high-order dependencies.

**Compliance With Llm Reviewing Policy:**

Affirmed.

**Final Justification:**

I have read the rebuttal and adjusting my score accordingly.

**Key Questions For Authors:**

- Please see the weakness.
- Also, the authors introduce the term "graph-structured brain representation." However, the final output appears to be an adjacency matrix ($A$), which is the standard format for a brain network. Could the authors clarify whether this "representation" refers to something beyond a learnable adjacency matrix? If the core contribution lies in the end-to-end learning of the network structure, it might be more accurately described as a "learnable brain network construction" process. Please provide a clear justification for the use of the term "graph-structured representation" and explain how it differs conceptually from existing learnable connectivity methods.

**Limitations:**

Although the proposed representation captures high-order dependencies, it still focuses on pairwise relationships between ROIs. In practice, brain function often arises from coordinated interactions among multiple regions rather than purely pairwise connections. Extending the framework to capture multi-region functional connectivity  could further improve its ability to model complex brain network dynamics.

**Strengths And Weaknesses:**

Strength:
- (S1) This work shifts the focus from merely designing better predictors to the learnable construction of the brain network itself. By incorporating learnable high-order dependencies, the paper offers more flexible and expressive paradigm for neuroimaging analysis compared to static methods.

- (S2) The inclusion of theoretical proof showing that the framework can approximate complex dependencies adds credibility to the proposed method and distinguishes it from purely heuristic graph-learning approaches.

Weakness:
- (W1) The authors claim that previous works primarily focus on fixed brain networks and emphasize that their proposed method is unique in making the graph construction process learnable. However, the concept of end-to-end learnable graph construction for brain functional connectivity is not entirely novel and has been explored in prior literature, such as "BrainGNN" (Mahmood et al., 2021) and "AGMGC" (Noman et al., 2025). The manuscript fails to discuss these closely related works or provide a comparative analysis, which makes it difficult to assess the specific technical advancements of the proposed high-order dependency modeling over existing learnable graph approaches.

- (W2) Key experimental details, including the dataset descriptions, baseline configurations, and evaluation metrics, are only provided in the supplementary material. This hinders the self-containment of the main manuscript and makes it difficult for readers to fully assess the experimental setup.

- (W3) Figure 9 (circular visualization) shows duplicate ROI labels (e.g., "Insula_R", "Lingual_L"). This may be potential errors in the preprocessing pipeline or visualization. The authors should verify the ROI naming scheme and ensure that each ROI is uniquely identified to avoid confusion.

- (W4) Some abbreviations (e.g. , SEN, SPE, FC) are used in the main text without prior definition and are only explained in the supplementary material. Defining these abbreviations at their first occurrence in the main text would improve readability.

- (W5) In line 318, the authors claim "2.4%" ACC improvement over BQN (75.68 vs. 77.82). However, this reflects an increase of 2.14 percentage points, not a 2.14% relative improvement. The authors should revise the wording for accuracy.

[1] Mahmood, Usman, et al. "Attend to connect: End-to-end brain functional connectivity estimation." In ICLR 2021 Workshop on Geometrical and Topological Representation Learning  (2021)

[2] Noman, Fuad, et al. "Adaptive Graph Learning with Multi-graph Convolutions for Brain Disorder Classification." In MICCAI (2025)

---

> ### Author Rebuttal · Authors · 2026-03-31
>
> > W1& Key Question. The manuscript fails to discuss these closely related works or provide a comparative analysis, which makes it difficult to assess the specific technical advancements of the proposed high-order dependency modeling over existing learnable graph approaches.
>
> R1. Thanks for your insightful question. We want to emphasize that "graph-structured brain **representation**” is very different from “learnable graph (brain network) **construction**”, although both of them produce adjacency matrices. As shown in Fig. 1, **representation should simplify downstream predictors** by capturing rich semantic information (Lines 067-070 Right), rather than relying on flexible models to process raw data. The graph-structured brain representation learned from the proposed BRep is directly fed into the **simple MLP** (Line 216 Right) for predictions, since it captures rich information on the brain. This also facilitates the model interpretability. On the contrary, all existing learnable connectivity methods, including BrainGNN and AGMGC, require complex models, e.g., GNNs and Graph Transformers, to process the constructed brain networks. This means the constructed brain networks are only raw intermediate data and lack sufficient semantic information. Therefore, as noted in the first contribution (Line 094 Left), we believe that **graph-structured brain representation learning is a novel problem** beyond the scope of learnable brain network construction algorithms.
>
> Even so, we appreciate your suggestion and will add a related work section to discuss existing methods on end-to-end learnable graph construction for brain functional connectivity. We hope this will make the contributions of our paper clearer.
>
>
>
> > W2&W4&W5: Presentation and Clarity
>
> R2. According to your suggestions, we will address these issues as follows: a condensed version of core experimental setups will be moved from the supplementary material to main text Section 3 with optimized formatting to fit the page limit; abbreviations SEN, SPE and FC will be explicitly defined at their first occurrence; and the imprecise percentage wording will be corrected to an absolute increase of 2.14 percentage points for accuracy. Additionally, we will carefully proofread the entire manuscript to ensure consistency and precision.
>
>
>
>
>
> > W3. Duplicate ROI labels in Fig. 9.
>
> R3. This is strictly a visualization flaw in the plotting script, not a data preprocessing error. The underlying matrices are strictly computed using the full ROIs from the Craddock-200 atlas. To prevent visual overlapping, the original plotting script applied a dense-filtering mechanism (displaying a subset of ~100 nodes) and mapped these fine-grained ROIs to their macro-anatomical AAL labels (e.g., "Insula_R"). Since multiple fine-grained ROIs belong to the same macro-anatomical region, this mapping inadvertently resulted in visually duplicate labels.
>
> To rigorously correct this and ensure the biological findings are accurately represented globally, we adopt the protocol utilized in MCST-GCN [1] to update the visualization (shown in **[Results](https://anonymous.4open.science/r/BRep-demo-1A3E/fig/chord.png)**), where the ROIs in the same area are merged in the chord diagram for a clearer visualization.
>
>
>
> [1] Spatio-temporal graph hubness propagation model for dynamic brain network classification. IEEE Transactions on Medical Imaging, 2024.
>
>
>
>
>
> > Key Question. Please provide a clear justification for the use of the term "graph-structured representation" and explain how it differs conceptually from existing learnable connectivity methods.
>
> R5. Please refer to the **Response to W1**.

---

> > ### Author Rebuttal · Reviewer_MjJS · 2026-04-03
> >
> > Thanks for the clarification.
> > I have no further questions.

---

### Decision · Program_Chairs · 2026-04-30

**Decision:**

Accept (regular)

**Comment:**

This paper addresses the limitations of traditional hand-crafted brain functional networks by proposing a learnable graph-structured representation. Reviewers agree that this provides a more flexible alternative to traditional functional connectivity estimation pipelines that are typically decoupled from downstream objectives and that the theoretical analysis provided is interesting and relevant. Although there are some concerns on the novelty, all reviewers's final scores remains positive. So I recommend acceptance.